# Formulation, *in vitro* characterization and optimization of taste-masked orally disintegrating co-trimoxazole tablet by direct compression

**Chernet Tafere**[1]*, **Zewdu Yilma**[1], **Solomon Abrha**[2], **Adane Yehualaw**[1]

**1** Department of Pharmacy, College of Health Sciences, Bahir dar University, Bahir Dar, Ethiopia,
**2** Department of Pharmaceutics, School of Pharmacy, College of Health Sciences, Mekelle University, Mekelle, Ethiopia

* cheru004@gmail.com

**Data Availability Statement:** All relevant data are within the manuscript and its Supporting Information files.

## Abstract

### Introduction

Orally disintegrating tablet (ODT) is a dosage form that overcomes the problem of swallowing which is prevalent in about 35% of the general population. Co-trimoxazole (CTX) is given for patients with HIV for the prophylaxis of opportunistic infection (OI), commonly for pneumocystis carinii pneumonia. It was reported that CTX was associated with a 25–46% reduction in mortality among individuals infected with HIV in sub-Saharan Africa. Esophageal candidiasis which usually comes along with HIV/AIDS is one of AIDS defining illness affecting up to 1 in 5 of people with AIDS. This opportunistic illness is manifested by painful or difficulty of swallowing. In this respect, CTX ODT offer the advantages of both liquid dosage forms in terms of easy swallowing thereby improve patient compliance and solid dosage forms in terms of dose uniformity, stability, lower production, and transportation costs. The objective of this study was to formulate, characterize and optimize CTX ODT which could overcome swallowing problem and improve patient compliance. Co-trimoxazole ODTs were prepared by direct compression technique using a semi synthetic super disintegrant (crospovidone) along with other excipients. Two taste masking techniques were employed, addition of sweetening agent, and solid dispersion by using a pH sensitive polymer, Eudragit E-100 at different ratios (1:1, 1:2 and 1:3). Taste masking was determined by comparing taste threshold value and *in vitro* drug release. Preliminary study was used to investigate the effect of crospovidone, compression force (CF) and Hydroxypropyl cellulose (HPC) on disintegration time, friability and wetting time (WT). Factorial design was used as it enables simultaneous evaluation of formulation variables and their interaction effect. From the preliminary study, the factors that were found significant were further optimized using central composite design. Design-Expert 8.0.7.1 software was employed to carry out the experimental design. The bitterness threshold concentration of Trimethoprim was found to be 150 µg/ml and the *in vitro* drug release of the three batches of drug to polymer ratio (F1:1, 1:2 and 1:3) was 2.80±0.05, 2.77±0.00 and 2.63±0.00 respectively. From the optimization study, the optimal concentration for the superdisintegrant was 8.60% w/w and a CF of 11.25

**Funding:** This research was financially funded by Adigrat University, Adigrat, Ethiopia, for the partial fulfillment of a master's degree in pharmaceutics at Mekelle University. In addition to Adigrat University, Addis Pharmaceutical Factory provided the materials including the active pharmaceutical ingredients and excipients freely. The funders had no role in the study design, data collection and analysis, decision to publish, or preparation of the manuscript. No authors received any salary from the funders.

**Competing interests:** The authors have declared that no competing interests exist.

KN which gave a rapid disintegration and WT of 13.79 and 23.19 seconds respectively and a friability of 0.666%.

## Conclusion

In this study, co-trimoxazole ODT was formulated successfully. Central composite design was effectively used to model and optimize friability, DT and WT. The method was found effective for estimating the effect of independent variables on the dependent variables by using polynomial equation and surface plots. Optimization of the response variables was possible by using both numerical and graphical optimization and the predicted optimal conditions were confirmed experimentally and were found to be in good agreement within 5% of the predicted responses. The results of the study showed that CTX ODT had significantly rapid disintegration, less than 1% friability and enhanced dissolution profiles. The successful formulation of CTX ODT can solve difficulty of swallowing of conventional tablets for some group of patients which are unable to swallow solid oral dosage form.

## 1. Introduction

Oral route is the preferred method of drug administration for systemic effect, and tablets are the most popular of all dosage forms existing today due to their convenience of self-administration, ease and accuracy of dosing, and most importantly patient compliance [1, 2]. Besides, as solid oral delivery systems do not require sterile conditions, they are less expensive to manufacture and provide better stability for the drugs as compared with liquid dosage forms [3, 4]. For these reasons most therapeutic agents considered for systemic drug delivery have a tendency to be administered via the oral route [5]. But, the oral tablets are associated with some problems; particularly for those segments of the population which are unable to swallow solid oral dosage forms. The most vulnerable groups under this category are pediatric patients: because of underdeveloped muscular and nervous control and geriatric patients as a result of changes in various physiological and neurological conditions associated with aging [6, 7]. However, difficulty in swallowing or dysphagia was also reported to affect nearly 35% of the general population. It is also a pertinent finding with a number of pathological conditions, including stroke, parkinson's disease, neurological disorders and AIDS [8]. In addition to dysphagic patients, those who are traveling with no or little access to water are similarly affected, potentially limiting patients not to take orally administered conventional tablets or capsules [9].

In a nutshell, difficulty of swallowing appears to impact patients compliance or adherence to their prescribed medication [10]. With the aim of facilitating the ease of oral medication administration and increasing patient compliance, researches have been directed at developing innovative dosage forms for oral administration. Among the dosage forms developed for this purpose, the ODTs have been considered by product development scientists as the best solution for patient suffering from dysphagia [11].

ODT is a solid dosage form containing a medicinal substance which is designed to disintegrate within 30 seconds or less when placed upon the tongue and get in contact with saliva without the need of water [12].

ODTs offer the advantages of both solid dosage forms and liquid dosage forms [5]. They provide better stability for drugs as conventional tablet formulations and they are easy to swallow like the liquid dosage forms [13, 14]. Because of the ability of ODTs to present solids in the form of suspension or solution even when placed in the mouth under limited bio-fluid without the

need of water, it is found to be crucial for busy people who do not always have access to water [15]. Besides, ODTs avoid the risk of choking (Physical obstruction) that are usually encountered during administration of conventional oral formulations, thus providing improved safety [16].

The ODTs are far better than the oral liquid dosage forms in terms of accurate dosing and easy handling by patients [8]. In addition, they are easy to transport and bring less handling and transportation costs due to their small volume and weight. Their production costs are less, which makes them more affordable than standard liquid formulations [13, 17].

Co-trimoxazole (CTX) is a fixed-dose combination of two antimicrobial drugs namely, sulfamethoxazole (SMX) and trimethoprim (TMP) in 5:1 ratio. TMP is an odorless white powder with a bitter taste. [18]. SMX on the other hand is a white to off-white crystalline powder [19]. Based upon the biopharmaceutical classification system both TMP and SMX are class II drugs; which implies low aqueous solubility and high membrane permeability [20].

CTX is a broad-spectrum antimicrobial agent used to treat the urinary, respiratory, and gastro-intestinal tract infections [21, 22]. The antibiotic is categorized as a vital medicine and is being used in HIV programs. It is included in the opportunistic infection (OI) standard treatment guideline (STGs) and the general treatment guidelines of Ethiopia. It is also included in all three editions of the essential medicines list of Ethiopia [23].

Besides, owing to its prophylactic effect on several secondary infections in people living with HIV/AIDS such as pneumocystis carinii pneumonia (PCP), WHO incorporates CTX as an integral component of the HIV chronic care package and many countries including Ethiopia are using this drug for this purpose [24, 25]. When it is taken regularly as prophylaxis, it reduces mortality and specifically lowers the risk of OIs in adults and children living with HIV even in resource limited areas [23]. For instance, in one study in sub-Sahara Africa, it showed a 25–46% reduction of mortality among individuals infected with HIV even in areas with high bacterial resistance to the antibiotic [26]. However, esophageal candidiasis which is on the list of AIDS defining illness, affect up to 1 in 5 AIDS patients and are responsible for painful or difficulty of swallowing [27]. The dysphagia associated with this comorbidity can affect patients compliance to treatment regimen including CTX given for these patients.

Observing on the impact of difficulty of swallowing in children, elderlies and HIV/AIDS patients, various stake holders recommend developing orodispersible tablets. For instance, in 2008, the WHO experts proposed to formulate ODTs for children age of ≤5 years and recommended to take various climate zones into consideration during formulation [28]. In this vein, significant number of drugs has been formulated in the form of ODTs (Ibuprofen, Jadhav *et al*. 2016; Fluoxetine, Ali *et al*. 2016) but co-trimoxazole is not one of them.

CTX is the treatment of choice for typhoid fever in the pediatric patient population particularly in resource limited areas like Ethiopia. This is due to the fact that the first line treatment (ciprofloxacin) is contraindicated in children under 18 because of joint/cartilage toxicity. As children are the major group of people who have difficulty of swallowing solid oral dosage forms, formulation of CTX ODT could help improve compliance.

Due to the above mentioned drawbacks, CTX ODT is expected to be best alternative medication for patients who are unable to swallow conventional CTX tablets either due to age related factor or disease related one. As a result, the present study aimed to formulate and optimize taste-masked orally disintegrating co-trimoxazole tablet using direct compression.

## 2. Materials and methods

### 2.1 Materials

Trimethoprim (Unival Chem.ltd, China) (RV062), sulfamethoxazole (Hubei XinRunde chemical Co; Ltd, china) (RS143), saccharin sodium (Tianjin Zhongrui pharmaceutical co; Ltd,

China) (RS165), colloidal silicon dioxide (RA168) and crospovidone (RP153) (Boai NKY, Pharmaceutical Ltd./ India), Hydroxypropyl cellulose, Eudragit E100 and magnesium stearate (RM190) (Anhui Menovo Pharmaceutical Co; Ltd, China), mannitol (RM135), MCC PH 102 (RM144) (China Associate Co. Ltd, China), and sodium lauryl sulfate (Sinolight chemicals Co; Ltd, China) (RS166), were kindly donated by Addis Pharmaceutical Factory (APF). HCl (Loba Chemie Pvt. Ltd., Mumbai, India), Methanol, Triethylamine and acetonitrile HPLC grade (Sigma Aldrich Chem Pvt. Ltd., bengaluru, India) were all purchased from local market.

## 2.2 Methods

**2.2.1. Taste masking by solid dispersion method.** Taste masking was performed according to the method described by Sharma *et al.*, (2010). Different ratios (1:1, 1:2, and 1:3) of TMP (bitter drug) and powdered Eudragit E-100 were mixed using mortar and pestle. The mixtures were transferred into a stainless steel vessel and 10% ethanol (10 mL) was added to each mixture (drug polymer blend). Each mixture was then stirred constantly using a magnetic stirrer until a thick gel was formed. Incorporated ethanol was removed by evaporation at 40˚C in a thermostat oven for 8 hours, and the solidified gel was then crushed into particles using mortar and pestle. Finally, each crushed drug -EudragitE100 blend were allowed to pass through a 45 mesh sieve and stored appropriately (in a closed cabinet to protect it from direct sun light exposure) for further use. In addition to Eudragit E100, the sweetener saccharin sodium was also employed as a taste masker at 1% w/w while formulating CTX ODT.

*2.2.1.1. Selection of best taste masked drug-polymer blend.* The best taste masked candidate was selected based on taste threshold concentration, which is the minimum concentration among a range of dilutions of a substance at which the volunteer just starts feeling the bitter taste [29].

*2.2.1.2. Determination of taste threshold value.* The threshold of bitterness concentration for the solid dispersion was determined as per the method described by [30]. The research was done in the year 2017 up to 2018. A panel of 6 (3 males and 3 females) healthy, non-smoking, literate, with the age >18 years, and who are voluntary to participate in the study were taken. The participants were recruited by purposeful sampling technique in Mekelle University, Mekelle Ethiopia. Prior to the commencement of taste masking evaluation, ethical clearance from Mekelle University Health Sciences College Ethical Review Committee (MUHSCERC) and informed consent from participants was taken. The aim of the study, related consequences of the study, and the confidentiality of their information was made clear to the study participants.

The following procedure was followed to determine the bitter taste threshold concentration/value of TMP [29]. Different concentrations of the drug samples (100, 150, 200, 250 and 300 μg/ml) were prepared in phosphate buffer pH 6.8. The study participants were requested to place 1mL of each sample (starting from the lowest to highest concentration) at the middle of their tongue and spit out after 30 seconds. There was a 10 minute interval before they took the next higher dose. The smallest concentration at which volunteers felt the bitter taste was recorded and taken as taste threshold value.

*2.2.1.3. In vitro release test at salivary pH.* The drug release from the taste masked granules was determined by *in vitro* test evaluation at salivary pH [29]. TMP- Eudragit E100 complex containing equivalent to 20 mg of TMP was placed in 10 mL of phosphate buffer pH 6.8 and shaken for 30 seconds. The amount of drug released was then analyzed by using HPLC (Agilent 1260 infinity, USA). The ratio of drug polymer complex that yielded drug release values just below the taste threshold concentration was selected to be used for taste masking purpose.

**Physicochemical characterization of the powder blend**

**Density related properties**

Density related properties were measured according to USP30-NF25 (2007).

**Bulk and tap density** were measured by pouring 30 g of powder blend into 250 mL dry graduated cylinder according to Eqs (1) and (2).

$$D_b = \frac{Mass\ of\ the\ powder\ (M)}{Volume\ of\ the\ powder\ (Vo)} \qquad \text{Eq (1)}$$

$$Tapped\ density = \frac{Weight\ of\ powder\ blend\ (M)}{Tapped\ volume\ of\ powder\ blend} \qquad \text{Eq (2)}$$

Where, Db is the bulk density (g/mL) M: mass of the powder blend (g) and Vo is the bulk volume

**Carr's index and Hausner's ratio** were also calculated from the bulk and tapped densities of the powder to assess flowability according to Eqs (3) and (4) respectively.

$$CI = \frac{tapped\ density - bulk\ density}{tapped\ density} \times 100 \qquad \text{Eq (3)}$$

$$HR = \frac{tapped\ density}{bulk\ density} \qquad \text{Eq (4)}$$

Where, CI is Carr's index and HR Hausner's ratio.

**Angle of repose**:

The angle of repose was determined from the dimensions of the powder pile, which was formed when 30 g of powder mass was allowed to flow through a funnel. The orifice of the funnel was fixed at 2 cm top of the powder pile as it is being formed in order to minimize the impact of falling on the tip of the cone. The blend was then allowed to flow through the funnel freely onto the surface. The diameter of the formed powder cone was measured and the angle of repose was calculated from the formed powder pile by using Eq (5)

$$\theta = tan^{-1}\left(\frac{H}{R}\right) \qquad \text{Eq (5)}$$

Where, H = is height of the formed pile, R = radius of the pile

**2.2.2. Preparation of ODTs.** Orally disintegrating co-trimoxazole tablets were prepared by direct compression (DC) method. All the ingredients were allowed to pass through 45 meshes or (354 μm) sieve separately. All except magnesium stearate and colloidal silicon dioxide were mixed in a polybag for five minutes. The blends were then lubricated with magnesium stearate and colloidal silicon dioxide and further mixed in a polybag for three minutes. Finally, each batch of the blend which was ready for compression was converted into tablets using 10 mm flat round punch. The steps for manufacturing of co-trimoxazole ODT tablet by the DC method were as follows:

Weighing → Sieving → Blending → Lubrication → Compression

*2.2.2.1. Evaluation of tablets*. The prepared tablets were evaluated for the following characteristics.

*2.2.2.2. Friability testing*. Pre-weighed sample of 20 tablets was placed in a friability tester (ERWEKA, TAR20, Germany) and rotated at 25 rpm for 4 minutes. The tablets were then dedusted and reweighed, and the friability percentage was computed according to the formula

given below.

$$\text{percentage friability} = \frac{(initial\ weight - final\ weight) \times 100}{initial\ weight} \qquad \text{Eq (6)}$$

*2.2.2.3. Hardness test.* The hardness of 6 tablets from each sample was determined by using hardness tester (CALIVA, THT2, England). The tablets were placed in the space provided and the crushing strength (in kilo Newton) that caused each tablet to break was recorded.

*2.2.2.4. Tablet thickness.* The thickness of six tablets from each sample was also measured using hardness tester (CALIVA, THT2, England).

*2.2.2.5. Wetting time.* Wetting time test was conducted as per the method described by Tabbakhian *et al.* [31] and Kumar [8]. A piece of double folded tissue paper was placed in clean and dry petri dish plates containing 6 mL of water. Six tablets from each formula were carefully placed individually on the paper and the time elapsed to completely wet or the time taken by the water to reach the upper surface of the tablet was taken as the wetting time (WT).

*2.2.2.6. Disintegration time.* Disintegration time test was carried out according to (USP, 2007). Six tablets from each formulation were randomly selected and placed in a disintegration tester (GBCaleva Ltd., Model; DIST2, England) filled with 900 ml distilled water and maintained at 37 ± 2 °C. The time required for complete disintegration of the tablets with no palpable mass remaining in the apparatus was recorded as the disintegration time (DT).

*2.2.2.7. Calibration curve.* Stock solution of SMX and TMP was prepared by transferring 100 mg of SMX and 20 mg of TMP reference standards to a 100 mL volumetric flask. Methanol (50 mL) was added and sonicated for 5 minutes, and then diluted to volume. From this stock solution, 5 mL was taken and transferred to a 100 mL volumetric flask and diluted with the mobile phase (80% distilled water and 20% acetonitrile HPLC grade). From the second stock solution four different volumes (20, 15, 10 and 5 mL) of the solution were transferred to 25 mL volumetric flasks and diluted with the mobile phase. The peak area readings of the second stock and the four solutions prepared from the second stock solution were made at 254 nm using HPLC (Agilent 1260 infinity, USA). The peak area versus concentration of solutions were plotted to obtain the calibration curve.

*2.2.2.8. Dissolution studies.* An *in vitro* dissolution study for each batch of tablets was conducted using USP Apparatus II, paddle method, (Pharma test, PTWS, Germany). The paddle was adjusted to rotate at 75 rpm. HCl (0.1N) (900 ml, pH 1.2), maintained at 37 ± 0.5°C was used as the dissolution medium [32]. Samples of dissolution medium (5 mL each) were withdrawn at predetermined time intervals (5, 10, 15, 30, 45 and 60 min) and immediately replaced with an equal volume of fresh dissolution medium (maintained at 37 ± 0.5°C) in order to maintain constant volume in the dissolution vessels. The samples withdrawn were filtered, properly diluted with mobile phase, and analyzed for TMP and SMX content in a validated HPLC method. The percentage of each active component dissolved was calculated by comparing of the peak responses obtained from a filtered aliquot of the solution under test with the peak responses from the corresponding component obtained from the standard preparation (USP, 2007).

*2.2.2.9. Drug content determination.* Drug content was determined by following the method described in (USP, 2007). The assay was prepared by weighing finely powdered 20 tablets. An accurately weighed portion of the powder, containing equivalent to 160 mg of SMX and 32 mg of TMP, was transferred to a 100-mL volumetric flask. Methanol (50 mL) was added and the mixture was then sonicated, with intermittent shaking, for 5 minutes. The solution was equilibrated to room temperature, diluted with methanol to volume and mixed thoroughly, then

filtered. After filtration, 5 ml of clear filtrate was transferred to a 50 mL volumetric flask, and diluted with the mobile phase to volume.

Standard preparation (20 μL) and the prepared assay were separately injected into the chromatograph, the chromatograms were recorded, and the responses for the major peaks were measured. The quantities (mg) of TMP and SMX were calculated by using the formula given below: (USP, 2007)

$$m = \frac{1000C(rU)}{rS} \qquad \text{Eq (7)}$$

Where C is the concentration (mg/mL), of the appropriate USP reference standard in thestandard preparation; and rU and rS are the responses of the corresponding analyte obtained from the assay preparation and the standard preparation, respectively.

**HPLC condition**

The amount of TMP and SMX dissolved in the dissolution study was determined by employing HPLC system. A sample of 5 mL was taken at each time interval from the dissolution medium and put into a volumetric flask (25 mL) and diluted to volume with the mobile phase. From these (20 μL) of the assay sample was injected into the chromatograph after being filtered through 0.45 μm membrane filter and were analyzed using HPLC (Agilent 1260 infinity, USA). All chromatographic analyses were carried out at 25°C. The compounds were separated using isocratic elution system. The percentage of each active component dissolved was calculated by comparing the peak responses of the test solution with the peak response of the corresponding standard preparation. The analysis was conducted as described in the (USP, 2007). The following chromatographic conditions were employed:

**Mobile phase:** Water (1400 mL), acetonitrile (400 mL) and triethylamine (20.0 mL) were mixed in a 2000 mL volumetric flask. After equilibrating to room temperature, the pH of the solution was adjusted with dilute glacial acetic acid (1 in 100) to make the pH in the range between 5.9 ± 0.1. Finally, the solution was diluted with water to volume, and filtered through a 0.45-μm membrane filter.

**Standard preparation:** Accurately weighed portion of the powder which equals to 32 mg TMP reference standard (RS) and 160 mg of SMX RS were dissolved in 100 mL methanol. From this solution (5 mL) was transferred to a 50 mL volumetric flask, and diluted with the mobile phase to volume, and mixed to obtain a standard preparation having known concentrations of about 0.032 mg/mL of TMP and 0.16 mg/mL of SMX.

**Chromatographic system:** The liquid chromatograph was equipped with a 254-nm detector and a 3.9 mm × 300 mm column that contains packing L1. The flow rate was 1.5 mL per minute.

**2.2.3. Experimental design for optimization of CTX ODT.** In this study, Design Expert Software (trial version 8.0.7.1) was used for data analysis. Before the optimization study, preliminary screening was carried out. In the preliminary study a 2 level (minimum and maximum) full factorial design was employed to evaluate the effect of 3 independent variables (Crospovidone and HPC concentration and compression force on the outcome variables (DT, WT and friability). The excipients which were used as fillers, namely mannitol and MCC pH102 in combination, were used to maintain a constant tablet weight of 260 mg. The other constituents and processing variables were kept constant. In full factorial design for 3 factors at 2 levels, a total of 8 experimental trials (i.e. $2^k$) combinations are found. The number 2 indicates the level and K, shows the number of factors or independent variables. Composition of the preliminary experiment and experimental levels are shown in Tables 1 and 2 respectively.

After the preliminary experiment, the factors that were found to be significant (CF and crospovidone concentration) were optimized by employing a CCD with five coded values as

**Table 1. Composition of tablet for the preliminary experiment.**

| Formulation ingredients | F1H | F2H | F3H | F4H | F5L | F6L | F7L | F8L |
|---|---|---|---|---|---|---|---|---|
| Trimethoprim | 20 | 20 | 20 | 20 | 20 | 20 | 20 | 20 |
| Sulfamethoxazole | 100 | 100 | 100 | 100 | 100 | 100 | 100 | 100 |
| Eudragit E100 | 20 | 20 | 20 | 20 | 20 | 20 | 20 | 20 |
| MCCPH102 | 43.25 | 32.85 | 39.35 | 28.95 | 43.25 | 32.85 | 39.35 | 28.95 |
| Mg stearate | 1.3 | 1.3 | 1.3 | 1.3 | 1.3 | 1.3 | 1.3 | 1.3 |
| Aerosil | 2.6 | 2.6 | 2.6 | 2.6 | 2.6 | 2.6 | 2.6 | 2.6 |
| Saccharin sodium | 2.6 | 2.6 | 2.6 | 2.6 | 2.6 | 2.6 | 2.6 | 2.6 |
| SLS | 2.6 | 2.6 | 2.6 | 2.6 | 2.6 | 2.6 | 2.6 | 2.6 |
| Mannitol | 57.25 | 46.85 | 53.35 | 42.95 | 57.25 | 46.85 | 53.35 | 42.95 |
| Crospovidone | 5.2 | 26 | 5.2 | 26 | 5.2 | 26 | 5.2 | 26 |
| HPC | 5.2 | 5.2 | 13 | 13 | 5.2 | 5.2 | 13 | 13 |

FH = Formulation at high compression force, FL = Formulation at low compression force, Average weight = 260 mg

shown in Table 2. For a 2 factor study in CCD, the total number of experiments to be performed in the design are generally given as sum of $2^n$ factorial runs, 2n axial runs, and $n_c$ center runs ($2^n + 2n + n_c$), where n is the number of factors. Therefore, for n = 2, the total number of experiments would be 13: ($2^2 + (2 \times 2) + 5$) five level for each factor [33]. The 13 experiments were carried out to find the optimum area, at which the desired responses were achieved.

*2.2.3.1. Statistical analysis.* Statistical analysis of all batches was performed with Microsoft Excel and plots of drug release profiles were constructed using Origin 8 Software (Origin Lab Corporation, MA, and USA). To demonstrate the influence of each factor on responses graphically and indicate the optimum level of factors, the contour and response surface plots were generated using Design-Expert 8.0.7.1 software. For the ANOVA, 95% confidence interval was chosen and the values were considered statistically significant when $p < 0.05$.

# 3. Results and discussions

## 3.1 Bitterness threshold concentration of co-trimoxazole

From the different concentrations of TMP standard solutions, four of the participants felt the bitterness at the concentration of 200 μg /mL; whereas, the remaining two participants felt it at 150 μg /mL. Therefore, 150 μg/mL was taken as the bitterness threshold concentration of TMP.

**3.1.1. *In vitro* taste-masking evaluation.** SD was the method used for taste masking. This technique, in which the drug is molecularly dispersed within the polymer matrix have shown effectiveness for masking of drugs with unpleasant taste [34]. In addition, SD with Eudragit E100, had showed better dissolution profile than the marketed formulations when diclofenac was used as a model drug [35]. Since TMP has poor aqueous solubility, this method was

**Table 2. Experimental levels of independent variables for optimizing CMX ODT formulation.**

| Variables | Levels | | | | |
|---|---|---|---|---|---|
| | -α | -1 | 0 | +1 | +α |
| CF ($X_1$) | 2.93 | 5 | 10 | 15 | 17.07 |
| Crospovidone ($X_2$) | 0.34 | 2 | 6 | 10 | 11.66 |

α = 1.41421

selected as it can be used for both taste masking and solubility enhancement. SD only masks the taste, but does not impart sweetness or palatability for the formulation. The sweetener in contrast provides further sweetness and palatability which can easily be taken by patients particularly pediatrics. For this reason the sweetener (Saccharin sodium) was used to augment taste masking. Among the formulations, formulation 1:1 (drug: polymer) was regarded the most cost effective formulation because its drug release was small as compared with bitterness threshold, and uses the minimum amount of the polymer accompanied with low cost of polymer as compared to the other formulations. Hence, it was incorporated as one formulation ingredient for the screening and optimization studies.

Drug release evaluation of the three batches of drug-polymer blends is presented in Table 3. According to the US FDA, ODT is designed to disintegrate rapidly, usually within 30 seconds [12]. For this reason, the *in vitro* test was determined by placing TMP- Eudragit E100 complex (1:1, 1:2 and 1:3) in 10 mL of phosphate buffer and shaken for 30 seconds. The phosphate buffer was used to simulate saliva's pH as the final formulated ODT is designed to disintegrate in the mouth. Accordingly, the concentration of drug released was far lower than the threshold concentration in all the three drug-polymer blends. Formulation (F) with 1:1 (drug: Eudragit E100) ratio released the maximum concentration of drug which was 2.80 μg/mL. This value is very small when compared with the bitterness threshold concentration (150 μg/mL) of the drug. This marked difference might be attributed to the nature of taste masker Eudragit E100, a pH sensitive polymer, which is insoluble at salivary pH (6.8–7.2) as well as in weakly acidic buffer solutions up to pH 5 [36]. In addition, a decrease in solubility of TMP as a function of pH was also reported by Sayar *et al* (2008), who observed a decrement of solubility from 154.1 to 55.1 mg/mL following a change in the pH of the solutions from 1.2 to 6.8.

## 3.2 Preliminary studies

Preliminary screening was carried out to identify the most critical variables that could have significant impact on the response variables. Based on the review of different literatures, super disintegrant concentration, binder concentration and CF were identified as factors. In the current study, CF and crospovidone were found as the two most significant factors which had an impact on DT, WT and friability.

For the production of tablets DC was employed as it has low cost of production, uses conventional equipment and commonly available excipients. In addition, it provides high mechanical integrity of tablets [8]. The popularity of DC has increased due to the introduction of superdisintegrants and a better understanding of their properties since the rate of disintegration and hence the dissolution is principally affected by them [37].

In order to formulate ODTs, superdisintegrants are the most required formulation ingredients. They are used at low level in solid dosage forms, typically 1–10% w/w relative to the total weight of the dosage unit [38]. There are different types of superdisintegrants in the market, such as, Croscarmellose sodium, sodium starch glycolate but crospovidone is the most commonly used superdisintegrant. It has highly porous particles which do not form complexes with drugs and pose no compatibility problem. Crospovidone has an efficient disintegrant

**Table 3. Drug release evaluation of the three batches of drug to polymer ratio.**

| Formulation code | Drug release (μg/mL) |
|---|---|
| F1:1 | 2.80±0.05 |
| F1:2 | 2.77±0.00 |
| F1:3 | 2.63±0.00 |

action at low concentration usually in the range of 2–5%. However at concentration up to 10%, it has a little effect on the flow properties of some other excipients and drugs [39]. Hence, in this study, 2% and 10% were selected as the low- level and high-level concentrations of the polymer, respectively.

A large number of efforts have been applied to modify conventional tableting formulation and/or the process in order to produce ODTs with rapid DT while maintaining sufficient mechanical strength. CF is known to affect tablet properties [40]. Increasing CF generally results in tablets with lower porosity and higher mechanical strength but longer DT. An optimum CF is required to combine these opposing characteristics [41]. Different literatures have used different ranges of CF to formulate ODTs. In a study aimed to investigate the effects of tablet diameter and CF on tablet hardness, DT and porosity, CFs of 10, 15 and 20 kN were studied and the 15 kN was found to be optimum [40]. In another study [42], ODT tablets were prepared by DC using CF from 5–25 kN and 15 kN was found sufficient to obtain tablets with acceptable hardness, friability and DT. Schiermeier and Schmidt (2002) also utilized low level (4 kN) and high level (10 kN), to investigate the effect of independent variables on ibuprofen ODT, WT and crushing strength as outcome variable. In this study, based on the aforementioned studies, the 5 and 15 kN were therefore utilized as the low and high level CF respectively, in order to maintain the balance between DT, WT and friability.

Binders are used in tablets to provide cohesive strength to powdered materials and can be added both in dry and wet form [43]. They improve unsuitable compression properties of most drugs and are used to obtain tablets with adequate strength and optimum friability. Binders should also show substantial plastic deformation. Pressure binders are employed in dry granulation and DC [44].

In this study hydroxypropyl cellulose (HPC) was used as a directly compressible binder. In formulation of tablets, HPC is primarily used in the concentrations range of 2–6% w/w in either wet-granulation, or DC processes [45]. Since tablet formulation with HPC was characterized by high plastic deformation and good compactibility [44], which is needed for direct compressible binders, 2 and 5% were employed as low and high level concentration respectively. Each of the preliminary batches was evaluated for pre-compression parameters and the results are presented below in Table 4.

**3.2.1. Powder characteristics of the preliminary formulations.** As displayed in Table 4, all formulations showed good to excellent flow property evidenced by density related data. From this density related data the compressibility index and the Hausner ratio was calculated and found in between 12.28 and 14.78 and 1.14 to 1.173 for CI and Hausner's ratio respectively. These results showed the good flow property of the powder to be formulated by DC. The flow properties of the powder blends were further analysed by determining the angle of repose for all the preliminary batches and the results were found in the range of 26.3 and 29.4 which further supports the good flowability of all the blends.

**3.2.2. Tablet characteristics of the preliminary formulations.** As shown in Table 5, the hardness of the tablets was in the range between 4.73 to 10.80 kgf. Weight variation was found

**Table 4. Physicochemical characterization of powder blends of the preliminary formulations.**

| Formulation code | Bulk density gm/ml | Tapped density gm/ml | Compressibility index | Hausner ratio | Angle of repose (°) |
|---|---|---|---|---|---|
| F1and F5 | 0.5±0.02 | 0.57±0.04 | 12.28±0.51 | 1.14±0.09 | 26.3±1.1 |
| F2 and F6 | 0.475±0.03 | 0.556±0.02 | 14.57±0.33 | 1.17±0.07 | 28.31±0.46 |
| F3 and F7 | 0.51±0.01 | 0.59±0.09 | 13.50±0.73 | 1.156±0.09 | 27.46±0.7 |
| F4 and F8 | 0.49±0.00 | 0.575±0.06 | 14.78±0.45 | 1.173±0.08 | 29.4±1.05 |

**Table 5. Post compression characteristics of the preliminary batches.**

| Codes | Thickness (mm) | Diameter (mm) | Hardness (Kgf) | Weight variation | Drug content (%) | |
|---|---|---|---|---|---|---|
| | | | | | TMP | SMX |
| F1H | 2.69±0.02 | 9.94±0.04 | 10.80±0.46 | 258.60±2.30 | 96.17±0.01 | 98.15±0.03 |
| F2H | 2.73±0.04 | 9.95±0.01 | 10.74±0.41 | 261.60±3.20 | 97.3±0.04 | 99.5±0.04 |
| F3H | 2.64±0.46 | 9.94±0.02 | 9.63±0.32 | 259.40±3.54 | 96.8±0.01 | 99.2±0.01 |
| F4H | 2.71±0.05 | 9.96±0.01 | 9.94±0.39 | 262.25±3.92 | 96.63±0.01 | 98.4±0.01 |
| F5L | 2.83±0.09 | 10.02±0.03 | 5.94±0.31 | 261.50±3.10 | 96.35±0.06 | 98.1±0.07 |
| F6L | 2.91±0.01 | 9.97±0.01 | 4.90±0.36 | 263.50±3.54 | 97.27±0.00 | 99.13±0.09 |
| F7L | 2.83±0.03 | 9.98±0.02 | 5.18±0.23 | 260.30±4.08 | 97.6±0.01 | 99.23±0.06 |
| F8L | 2.85±0.06 | 10.01±0.02 | 4.73±0.35 | 257.05±3.14 | 97.77±0.07 | 99.31±0.05 |

Weight variation = mean ± RSD All other results are mean ± SD, TMP = trimethoprim, SMX = sulfamethoxazole

to be in the acceptable range. For tablets with an average weight of 130–324 the acceptable range is below 7.5% of RSD (USP, 2007). Based on this, all the batches had fulfilled this specification. The content uniformity for Co-trimoxazole tablets according to (USP, 2007) necessitates not less than 93.0% and not more than 107% of the stated amount. Accordingly, all the preliminary batches passed the drug content specification.

**3.2.3. Effect of independent variables on response variables.** 3.2.3.1. *Effect of formulation variable on friability*. Achieving percentage friability within limits for an ODT is challenging to the formulator since all methods of manufacturing of ODT are responsible for increasing the percent friability values. Thus, it is necessary that this parameter should be evaluated [46].

Tablet formulations are required to meet the pharmacopoeial specification for friability, which is less than 1% weight loss (USP, 2007). The effect of the independent variables on friability is displayed in Table 6. As can be seen from this table, the friability values were in the range between 0.78% and 2.08%. These varying results of friability indicated that this outcome variable was strongly affected by one or more of the factors ($P<0.05$). In this case, it was significantly affected by the CF ($P = 0.0019$). When the CF was held at high value, the friability of the tablets became less than 1% while, those batches that were compressed at low level of CF failed the specification. It is generally known that an increase in CF results in an increase in tablet hardness, which in turn decreases tablet friability [40]. The other two factors didn't show any significant impact on the friability of the tablet ($P > 0.05$).

**Table 6. Results of response variables of the preliminary batches.**

| Formulation code | Friability (%) | Disintegration time (second) | Wetting time (second)(mean±Std) |
|---|---|---|---|
| F1H | 0.87 | 40 | 64±3 |
| F2H | 0.91 | 14 | 29±1 |
| F3H | 0.78 | 26 | 69±2 |
| F4H | 0.79 | 15 | 26±0 |
| F5L | 1.98 | 9 | 61±2 |
| F6L | 2.08 | 7 | 16±1 |
| F7L | 1.87 | 16 | 63±4 |
| F8L | 1.95 | 12 | 23±2 |

FH = formulation at high compression force, FL = formulation at low compression force

**Table 7. P values for all the three response variables.**

| Predictor variables | P values for the response variables | | |
| --- | --- | --- | --- |
| | Friability | Disintegration time | Wetting time |
| Compression force | 0.0019 | 0.173 | 0.0396 |
| HPC concentration | 0.2935 | 0.248 | 0.2560 |
| Crospovidone concentration | 0.4795 | 0.012 | < 0.0001 |

*3.2.3.2. Effect of formulation variable on wetting time.* The WT gives an insight about the disintegration property of the ODTs [47]. That is why it was incorporated as a factor for CTX ODTs. In addition, it mimics the action of saliva in contact with tablets [48].

WT was significantly affected with both crospovidone concentration and CF (Table 7) (P < 0.0001 and P < 0.05) respectively. When the CF was increased, the WT also increased and vice versa. For instance, when the CF was at high level (F2H), the WT was 29 seconds, but when the CF went to its low level keeping the other two constant (F6H) the WT was 16 seconds. On the other hand, crospovidone concentration had opposing effect on the WT. If we took F1H and F2H, we saw a decrement of WT from 64 to 29 seconds by increasing crospovidone concentration and keeping the other two constant.

*3.2.3.3. Effect of formulation variables on disintegration time.* From the results of the preliminary study, crospovidone was found to significantly affect the DT of the tablets (P < 0.05). When crospovidone concentration was taken from low to high keeping the other factors constant the disintegration decreased from 40 to 14 seconds (in F1H and F2H) and 26 and 15 (F3H and F4H) respectively.

As described in the above paragraph and depicted in Table 8, CF of and crospovidone concentration were the two statistically significant factors. As a result, they were selected for further optimization.

## 3.3 Optimization

After the important factors had been identified, the next step was to determine the settings for these factors that result in the optimum value of the responses. From the preliminary

**Table 8. Composition of the thirteen formulations.**

| Formulation code | Point type | Factors | |
| --- | --- | --- | --- |
| | | Compression force (KN) | Crospovidone Conc. (%) |
| F1 | Center | 10 (0) | 6 (0) |
| F2 | Axial | 10 (0) | 11.66 (+α) |
| F3 | Axial | 17.07 (+α) | 6 (0) |
| F4 | Center | 10 (0) | 6 (0) |
| F5 | Axial | 2.93 (-α) | 6 (0) |
| F6 | Factorial | 5 (-1) | 10 (+1) |
| F7 | Factorial | 15 (+1) | 2 (-1) |
| F8 | Center | 10 (0) | 6 (0) |
| F9 | Axial | 10 (0) | 0.34 (-α) |
| F10 | Center | 10 (0) | 6 (0) |
| F11 | Center | 10 (0) | 6 (0) |
| F12 | Factorial | 5 (-1) | 2 (-1) |
| F13 | Factorial | 15 (+1) | 10 (+1) |

Conc = concentration

experiment, CF and crospovidone concentration were identified as significant factors for further optimization. Thus their effect on friability, DT and WT were further studied using CCD.

The CCD was chosen because it is widely used for fitting a second-order model and it requires a minimum number of experiments to be performed. In CCD, the factorial points allow estimation of the first-order and interaction terms, axial points allow for efficient estimation of the pure quadratic terms and addition of the center points provide information about the existence of curvature in the design. Replicates of the center runs give a better estimate of the pure error and better power for the test. The number of replicates at the center point depends on the number of variables considered in the design and is generally between 3 and 10 [49, 50]. The CCD, for two independent factors generated 13 formulas (Table 8).

**3.3.1. Characterization of the powder blend.** The bulk density and the tapped density of the formulations ranged from 0.54 to 0.64 and 0.62 to 0.74 respectively. The Carr's index of the blends was between 11.00 to 16.13 and Hausner's ratio ranged from 1.12 to 1.19, showing good to excellent flow. Besides these values, the angle of repose was ranged from (21.6±1.63 to 33.2±1.34) supporting the good flow property of the powder blends showed in Carr's index Hausner's ratio. The powder characterization results are presented in Table 9.

**3.3.2. Characterization of tablets.** As shown in Table 10, the formulated tablets mean thickness and diameter values were found in the range of 2.54 to 2.98 mm and 9.91 to 10.01mm respectively. A decrease in tablet thickness was observed with increasing compression force (CF). The hardness of the tablets ranged between 2.96 to 11.47 kgf. This variance in hardness was attributed by the CF difference. In all the experiment as the CF increased the tablet hardness had also increased and vice versa. The assay result showed 95.8 ± 0.07 to 97.93 ± 0.06 for TMP and 97.73 ± 0.11 to 99.31 ± 0.05 for SMX which complied with the pharmacopoeial specification. The dissolution profiles of the optimized batches were also evaluated for their drug release pattern and the result is displayed in Figs 3 and 4.

*3.3.2.1. Calibration curve.* The peak area obtained for TMP and SMX were plotted against concentration (Figs 1 and 2 respectively). The peak responses were recorded by following the standard procedure (USP, 2007) and the relative retention times were 9.02 minute for TMP and 10.28 minute for SMX.

Quantitative estimation of drug sample was done by the calibration curves prepared in the concentration range of 10–50 μg/ml for SMX and 2–10 μg/ml for TMP. 20 μL of each dilution was injected three times into the column at a flow rate of 1.5 mL/min and the corresponding

**Table 9. Powder characteristics of the optimization experiment.**

| Formulation code | Bulk density g/mL | Tapped density g/mL | Compressibility index | Hausner ratio | Angle of repose |
|---|---|---|---|---|---|
| F1 | 0.63±0.08 | 0.72±0.07 | 12.50±0.43 | 1.14±0.02 | 31.4±1.21 |
| F2 | 0.61±0.07 | 0.73±0.08 | 16.13±0.29 | 1.19±0.03 | 33.2±1.34 |
| F3 | 0.61±0.03 | 0.72±0.03 | 15.31±0.63 | 1.18±0.08 | 29.0±1.87 |
| F4 | 0.59±0.05 | 0.69±0.04 | 15.45±0.37 | 1.18±0.02 | 32.5±1.1 |
| F5 | 0.61±0.01 | 0.71±0.02 | 14.29±0.69 | 1.17±0.04 | 28.14±0.89 |
| F6 | 0.63±0.06 | 0.72±0.07 | 13.54±1.20 | 1.16±0.07 | 24.3±2.26 |
| F7 | 0.61±0.04 | 0.70±0.04 | 12.64±0.34 | 1.14±0.01 | 25.4±1.21 |
| F8 | 0.54±0.03 | 0.62±0.03 | 13.75±0.59 | 1.16±0.01 | 27.3±0.68 |
| F9 | 0.60±0.00 | 0.67±0.00 | 11.00±0.00 | 1.12±0.00 | 21.6±1.63 |
| F10 | 0.60±0.04 | 0.70±0.03 | 14.06±0.63 | 1.16±0.04 | 32.6±0.74 |
| F11 | 0.64±0.01 | 0.74±0.02 | 13.83±0.86 | 1.16±0.06 | 26.5±1.58 |
| F12 | 0.61±0.06 | 0.70±0.06 | 13.93±1.10 | 1.16±0.05 | 32.6±0.87 |
| F13 | 0.60±0.09 | 0.70±0.09 | 13.71±0.51 | 1.16±0.04 | 28.7±2.14 |

**Table 10. Tablet characterization of the 13 optimized experiment.**

| Code | Thickness (mm) | Diameter (mm) | Hardness (kgf) | Drug content (%) | |
|---|---|---|---|---|---|
| | | | | TMP | SMX |
| F1 | 2.77±0.05 | 9.98±0.04 | 6.47±0.30 | 96.5±0.01 | 97.93±0.06 |
| F2 | 2.72±0.027 | 9.98±0.01 | 6.03±0.24 | 96.76±0.06 | 98.5±0.00 |
| F3 | 2.62±0.03 | 9.95±0.02 | 11.47±0.47 | 97.5±0.00 | 99.2±0.06 |
| F4 | 2.75±0.10 | 9.95±0.01 | 6.34±0.29 | 96.7±0.10 | 98.4±0.00 |
| F5 | 3.98±0.05 | 9.97±0.02 | 2.96±0.18 | 97.93±0.06 | 98.8±0.06 |
| F6 | 3.91±0.09 | 10.01±0.01 | 4.38±0.31 | 96.83±0.08 | 99.25±0.00 |
| F7 | 2.54±0.06 | 9.94±0.02 | 10.08±0.48 | 96.76±0.06 | 98.51±0.10 |
| F8 | 2.69±0.07 | 9.94±0.03 | 5.99±0.27 | 97.2±0.01 | 99.31±0.05 |
| F9 | 2.54±0.03 | 9.91±0.02 | 6.59±0.53 | 95.8±0.07 | 97.9±0.00 |
| F10 | 2.83±0.04 | 9.94±0.03 | 6.28±0.35 | 97.3±0.11 | 97.73±0.11 |
| F11 | 2.77±0.03 | 9.94±0.02 | 6.13±0.42 | 97.2±0.00 | 98.66±0.05 |
| F12 | 2.86±0.02 | 9.97±0.01 | 4.56±0.32 | 95.45±0.05 | 98.63±0.00 |
| F13 | 2.7±0.04 | 9.98±0.02 | 9.95±0.41 | 97.47±0.00 | 98.36±0.03 |

All the values are in the form of Mean ± SD

chromatograms were obtained. The calibration graph, constructed by plotting concentration of the drug against peak area showed a good linear regression equation of $Y_{tri} = 52.25x + 2.97$ and $Y_{sulfa} = 6970x + 69.05$ (Where Ytri and Y sulfa are peak areas of TMP and SMX respectively and x is the concentration of the two drugs). The coefficient of determination was found to be 0.9994 for SMX and 0.9996 for TMP.

*3.3.2.2. In vitro drug release.* The results of drug release profiles of TMP and SMX from different ODT formulations are illustrated in Figs 3 and 4 respectively. The *in vitro* drug release pattern was not evaluated for F5, F6 and F12 because they did not fulfill the pharmacopoeial specification for friability.

As shown in Figs 3 and 4, there was an immediate and superimposable release of both of the components. Formulations that disintegrate fast such as (B2- 6 seconds) showed fast

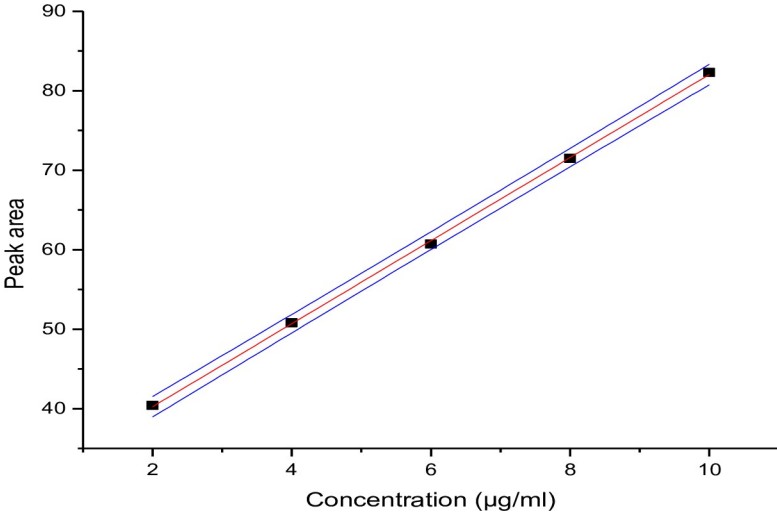

**Fig 1. Calibration curve of trimethoprim.**

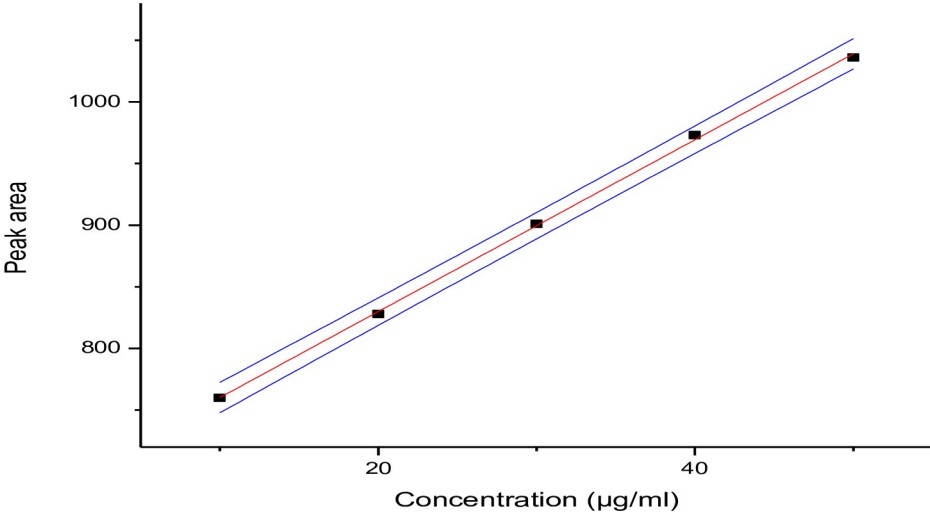

**Fig 2. Calibration curve of sulfamethoxazole.**

release of their content; 74.61% for SMX and 68.36% of TMP within 5 minutes. There was a small difference in their cumulative release of both components. All the ten batches showed a mean release of more than 70% of the stated amount within 60 minute; which conform to the pharmacopoeial specification. Since the rapid disintegration in the range of 6 to 57 seconds makes the drug available for the dissolution media, the difference in cumulative release among the different batches was negligible.

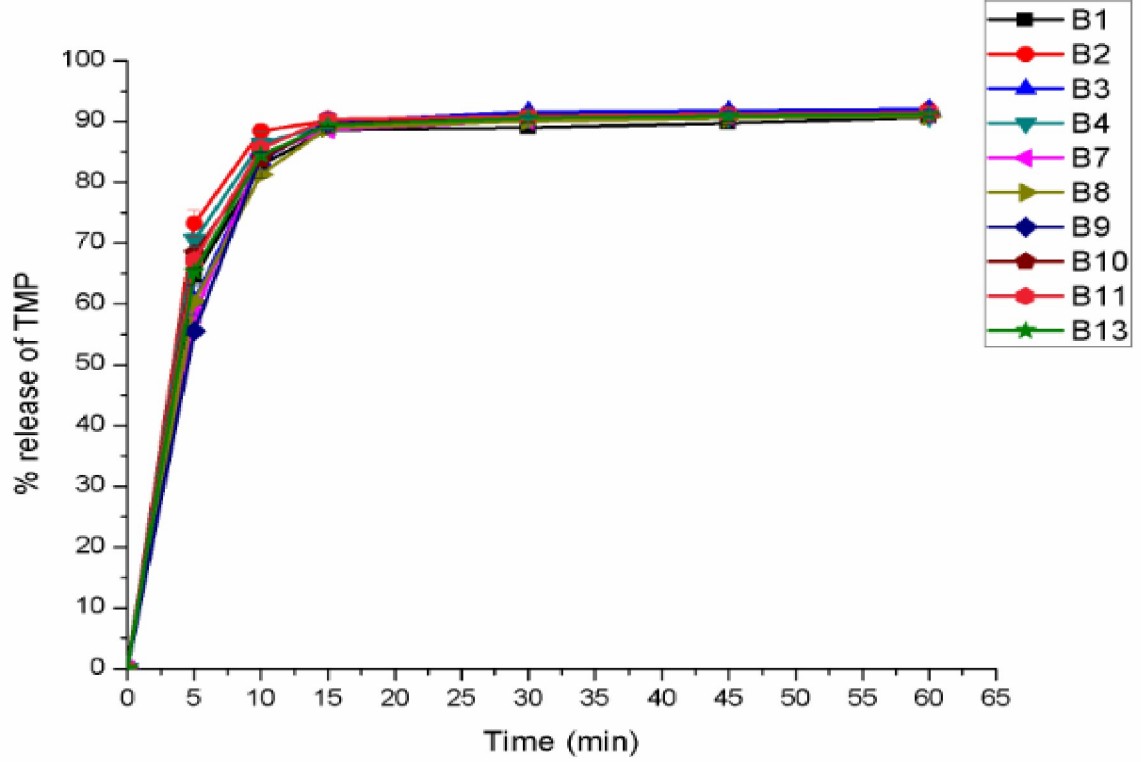

**Fig 3. *In vitro* drug release profiles of trimethoprim in the optimization batches of co-trimoxazole ODT tablets.**

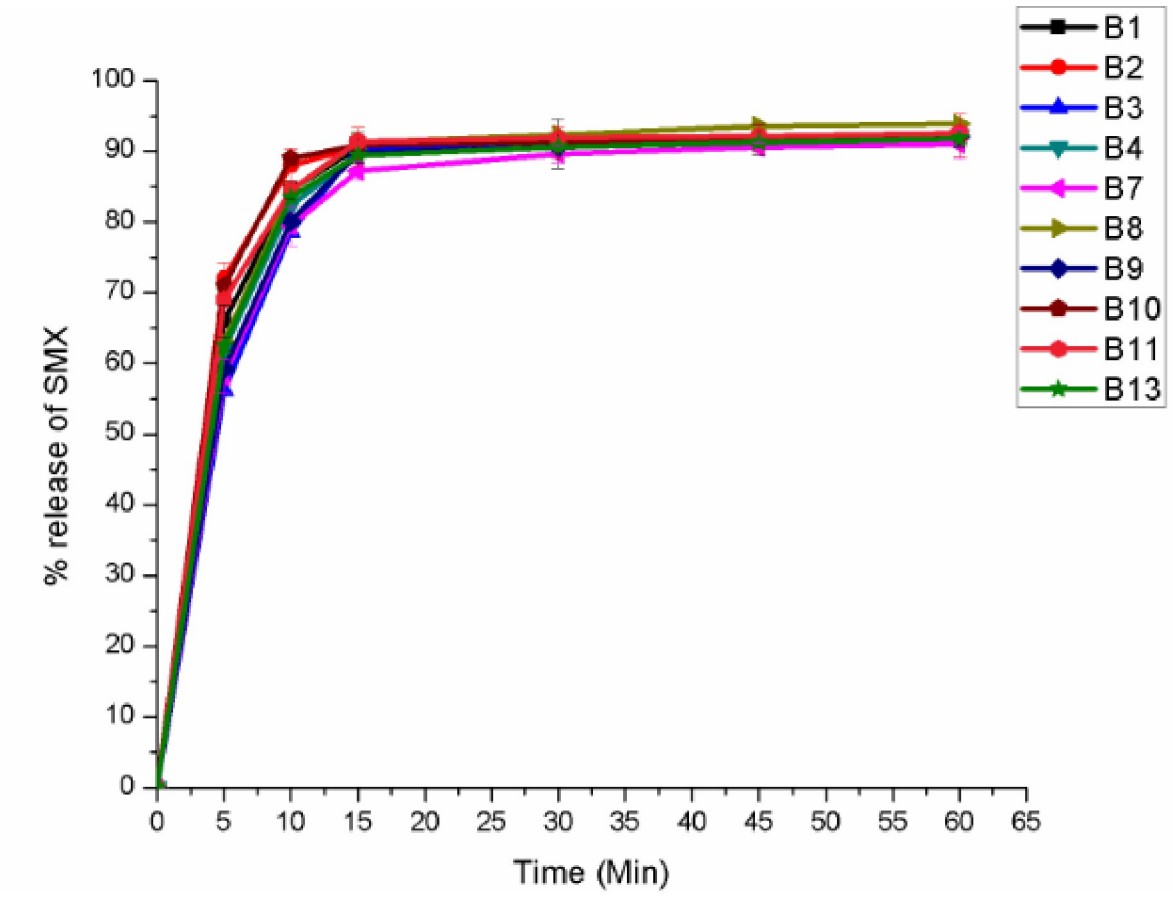

**Fig 4. *In vitro* drug release profile of sulfamethoxazole in the optimization batches of co-trimoxazole ODT tablets.**

*3.3.2.3. Evaluation of tablet for response variables.* Results of the response variables for the 13 optimization formulation of co-trimoxazole ODTs is displayed in Table 11.

**3.3.3. Model selection and checking adequacy.** It is always necessary to examine the fitted model to ensure that it provides an adequate approximation to the true system and verifies that none of the least square regression assumptions are violated. For checking model adequacy, several techniques can be considered. Model fitness and importance of individual terms can be specified by ANOVA. Terms with a P-value < 0.05 should be considered for inclusion. Lack of fit test is also important for choosing the best model. If a model shows a lack of fit, it should not be used to predict the response. An appropriate model should have insignificant lack of fit. Moreover, low PRESS and standard deviation, R-squared and predicted R-squared near one, are favorable [51].

For model selection different parameters were evaluated such as: reasonable agreement between adjusted R square and predicted R-square (within 0.20 of each other based on the software suggestion), not aliased, p-value of the model term which is less than 0.05, and the lack of fit p-value not-significant, meant to be greater than 0.05 [52]. The fit summary for the response variables is displayed in Table 12. Based on the fit summary, linear model was suggested for DT with a model P value of 0.0015 and lack of fit P value 0.0809 and the quadratic model was suggested for both friability, with a model P< 0.0001 and lack of fit P = 0.0590, and WT with a model P value and lack of fit P value 0.0035 and 0.0552 respectively. Hence, based

**Table 11. Results of the response variables for optimization formula of co-trimoxazole ODT.**

| Code | Responses | | |
|------|-----------|---|---|
| | Friability (%) | DT (Second) | WT (Second) (Mean ± SD) |
| F1 | 0.78 | 17 | 34.0±3.00 |
| F2 | 0.91 | 6 | 12.0±2.00 |
| F3 | 0.49 | 39 | 43.3±1.52 |
| F4 | 0.88 | 10 | 36±2.64 |
| F5 | 2.83 | 7 | 21.0±2.00 |
| F6 | 1.96 | 7 | 16.0±1.00 |
| F7 | 0.55 | 31 | 63.7±2.51 |
| F8 | 0.77 | 23 | 32.0±2.64 |
| F9 | 0.71 | 57 | 83.3±1.52 |
| F10 | 0.74 | 19 | 33±1.00 |
| F11 | 0.78 | 18 | 30.0±2.00 |
| F12 | 1.88 | 13 | 57.0±2.64 |
| F13 | 0.59 | 14 | 25.0±1.00 |

DT = disintegration time, WT = wetting time

on the fit summary output, linear model for DT and quadratic model for both friability and WT were selected as best fit models.

The goodness of fit of the model was also checked by coefficient of determination ($R^2$) (33). The R-squared values for friability and WT were found to be 0.9897 and 0.9785 respectively. These $R^2$ values indicate 98.97% and 97.85% of the variability in the response was explained with the models respectively. On the other hand R-squared value for DT was found to be 0.8912 which implies 89.12% of the variability in the response was explained by the model. The "Pred R-Squared" of 0.9373 is in reasonable agreement (within 0.2 for each other) with the "Adj R-Squared" of 0.9824 for friability and 0.8683 and 0.9632 for WT. This aggrement was also mainatained for DT with a value of 0.6744 for adjusted R-squared and 0.4910 for predicted R-squared.

**Table 12. Fit summary statistics for friability, wetting time and disintegration time.**

| Response | Source | R-Squared | Adjusted R¯Square | Predicted R- Square | PRESS | Remark |
|----------|--------|-----------|-------------------|---------------------|-------|--------|
| Friability | Linear | 0.7632 | 0.7158 | 0.5459 | 2.70 | |
| | 2FI | 0.7632 | 0.6843 | 0.4597 | 3.21 | |
| | Quadratic | 0.9897 | 0.9824 | 0.9373 | 0.37 | Suggested |
| | Cubic | 0.9981 | 0.9954 | 0.9949 | 0.03 | Aliased |
| WT | Linear | 0.8912 | 0.8694 | 0.7846 | 1074.40 | |
| | 2FI | 0.8915 | 0.8553 | 0.7302 | 1346.05 | |
| | Quadratic | 0.9785 | 0.9632 | 0.8683 | 656.91 | Suggested |
| | Cubic | 0.9961 | 0.9906 | 0.9874 | 62.62 | Aliased |
| DT | Linear | 0.8287 | 0.6744 | 0.4910 | 1423.49 | Suggested |
| | 2FI | 0.7635 | 0.6847 | 0.3726 | 1626.80 | |
| | Quadratic | 0.8462 | 0.7364 | 0.0974 | 2340.39 | |
| | Cubic | 0.9278 | 0.8267 | -1.4726 | 6411.38 | Aliased |

WT = wetting time, DT = disintegration time.

"Adeq Precision" measures the signal to noise ratio. A ratio greater than 4 is desirable and the model can be used to navigate the design space. In this respect, all the three responses had adequate precision greater than 4 with a value of 35.899 for friability 10.552 for DT and 26.263 for WT. On the other hand, PRESS is the sum of the squared differences between the experimental response and predicted response by regression model. It validates how this particular model fits each point in the design and can be used in the regression model selection [53].

The residuals also play an important role in judging model adequacy. It is calculated as ($e_i = y_i - \bar{y}$, i = 1; 2. . ...n; where, $e_i$, $y_i$ and $\bar{y}_i$ are the residuals of $i$th experiment, observed response and predicted response respectively). By constructing a normal probability plot of the residuals, a check for the normality assumption was made. If the residuals plot approximated along a straight line, the normality assumption is said to be satisfied. In addition, if the plot of studentized residuals versus predicted scattered randomly on the display, it suggests that the variance of the original observations was constant for all values of response which further confirms model adequacy [54].

As illustrated in Table 13, Values of "Prob > F" less than 0.0500 indicate model terms are significant. In this case $X_1$, $X^2_1$ are significant model terms with a P value of $< 0.0001$ for both the main and the quadratic effect. But crospovidone concentration ($X_2$), the interaction effect ($X_1X_2$) and $X^2_2$ are insignificant model terms. Therefore, backward elimination procedure was applied to reduce insignificant terms so as to increase the model's predictive efficiency.

Model term refinement should always be attempted before interpreting a satisfactory model. This process might increase both values of explained ($R^2$) and predicted variation. Refinement is primarily achieved through exclusion of the factors that are found to be insignificant in the coefficient plot [55].

After eliminating the insignificant model terms the ANOVA for friability was assessed and the result is depicted in Table 14. As displayed from the table below, the reduced model became more significant (F = 353.40, p < 0.0001) than the original model (F = 134.7, p < 0.0001) indicating the model predictive efficiency was improved. In both cases, the models were significant with (p < 0.0001) but the F-value was greater for the reduced model which specifies higher significance of the corresponding variable to cause an effect.

The reduced model also showed a slight improvement in the adjusted $R^2$ and high improvement for the predicted $R^2$ which went from 0.9373 to 0.9608 (Tables 12 and 15). a decreasement of predicted residual error sum of squares (PRESS), was also seen from 0.37 to 0.23.

According to Table 16 $X_1$, $X_2$, $X^2_2$ were significant model terms for WT. Where as, the interaction effects of $X_1$ and $X_2$, and the quadratic effect of of CF ($X^2_1$) on WT were insignificant model terms. As a result, model reduction was done to improve prediction efficiency.

**Table 13. Summary of ANOVA results of response surface models for drug friability.**

| Response | Source | Sum of squares | Df | Mean Square | F-value | P-value | Remark |
|---|---|---|---|---|---|---|---|
| Friability | Model | 5.88 | 5 | 1.18 | 134.7 | <0.0001 | Significant |
| | Compression force ($X_1$) | 4.51 | 1 | 4.51 | 517.02 | <0.0001 | Significant |
| | Crospovidone ($X_2$) | 0.020 | 1 | 0.020 | 2.32 | 0.1713 | |
| | $X_1X_2$ | 4.000E-004 | 1 | 4.000E-004 | 0.046 | 0.8366 | |
| | $X^2_1$ | 1.33 | 1 | 1.33 | 152.51 | <0.0001 | Significant |
| | $X^2_2$ | 1.087E-003 | 1 | 1.087E-003 | 0.12 | 0.7346 | |
| | Residual | 0.061 | 2 | 8.731E-003 | | | |
| | Lack of fit | 0.050 | 3 | 0.017 | 5.94 | 0.0590 | |
| | Pure error | 0.011 | 4 | 2.80E-003 | | | |
| | Core total | 5.94 | 12 | | | | |

**Table 14. Summary of ANOVA results of reduced quadratic model for friability.**

| Source | Sum of squares | DF | Mean square | F-value | P-value Prob > F | Remark |
|---|---|---|---|---|---|---|
| Model | 5.86 | 2 | 2.93 | 353.40 | < 0.0001 | Significant |
| $X_1$ | 4.51 | 1 | 4.51 | 544.59 | < 0.0001 | |
| $X_1^2$ | 1.34 | 1 | 1.34 | 162.21 | < 0.0001 | |
| Residual | 0.083 | 10 | 8.289E-003 | | | |
| Lack of fit | 0.072 | 6 | 0.012 | 4.27 | 0.0908 | Insignificant |
| Pure error | 0.011 | 4 | 2.800E-003 | | | |
| Cor total | 5.94 | 12 | | | | |

As shown from the reduced model ANOVA (Table 17) the P value and F value of the reduced model was more significant. Though the models' P-value for both reduced and un reduced models was the same (P< 0.0001), the F value changed from 63.75 to 132.98 which implies higher effect of each variable on the response. In addition The "Lack of Fit P-value" of 0.1084 implies the Lack of Fit is not significant and there is a 10.84% chance that a "Lack of fit" this large could occur due to noise. As we need the model to fit, a higher non-significant lack of fit (0.1084) is better than a lower non significant lack of fit (0.0552) since.

The adjusted $R^2$ and predicted $R^2$ had also shown improvement in the reduced formula. The adeqate Precision" went from 26.263 to 35.484 (Tables 12 and 18).

For DT, $X_1$ and $X_2$ were significant model terms with a P value of 0.0181 and 0.0015 respectively (Table 19). The "Lack of Fit F-value" of 4.59 implies there was 8.09% chance that a "Lack of Fit F-value" this large could occur due to noise.

In addition to the above parameters which were used for model selection and checking its adequacy, diagnostic checking tests enables researchers to evaluate adequacy of the models [56]. The normal probability plot of the residuals is used to check the normality assumption, if the assumption is true this plot will resemble a straight line. On the other hand a plot of the residual values versus the predicted response values is used to verify the absence of constant error. A random scattering of the residual values indicates that no correlation exists between the observed variance and the response [53, 57].

By applying the diagnostic plots provided by the software, such as normal probability plots of the studentized residuals, and the residuals versus the predicted, the model adequacy can be further confirmed [33]. Figs 5–10 shows the normal probability plots of the residuals and the plots of the residuals versus the predicted response for friability, DT and WT.

As shown from Figs 5, 7 and 9, the normality assumption was checked by the plot of normal probability versus studentized residual which showed cluster of the points around the straight line which underlines the errors are distributed normally for all the responses and the normality assumption was satisfied.

The assumption of constant variance was also tested through the plots of residuals versus the predicted response (Figs 6, 8 and 10). There was no obvious pattern and the residual was scattered randomly on the display. So, the assumption of variance homogeneity was satisfied in this work. Therefore, the models were reliable to describe and adequate for their respective responses.

**Table 15. The R-squared values of the reduced quadratic model for friability.**

| R-Squared | Adj R-Squared | Pred R-Squared | PRESS | Adeq Precision |
|---|---|---|---|---|
| 0.9860 | 0.9833 | 0.9608 | 0.23 | 51.431 |

**Table 16. Summary of ANOVA results of response surface models for drug wetting time.**

| Response | Source | Sum of squares | Df | Mean Square | F-value | P-value | Remark |
|---|---|---|---|---|---|---|---|
| Wetting time | Model | 4881.04 | 5 | 976.21 | 63.75 | < 0.0001 | Significant |
| | CF (X$_1$) | 281.35 | 1 | 281.35 | 18.37 | 0.0036 | Significant |
| | Crospovidone (X$_2$) | 4164.09 | 1 | 4164.09 | 271.95 | < 0.0001 | Significant |
| | X$_1$X$_2$ | 1.36 | 1 | 1.36 | 0.089 | 0.7746 | |
| | X$^2_1$ | 1.51 | 1 | 1.51 | 0.099 | 0.7625 | |
| | X$^2_2$ | 418.78 | 1 | 418.78 | 27.35 | 0.0012 | |
| | Residual | 107.18 | 7 | 15.31 | | | |
| | Lack of fit | 88.21 | 3 | 29.40 | 6.20 | 0.0552 | Insignificant |
| | Pure error | 18.98 | 4 | 4.74 | | | |
| | Core total | 4988.22 | 12 | | | | |

CF = compression force, DF = Degree of freedom

**3.3.4. The mathematical regression models.** In order to determine the levels of factors which yield optimum values of responses, mathematical relationships were generated between the dependent and independent variables. The final polynomial equations relating the dependent and independent variables in terms of coded factors were obtained as shown below from the regression analysis (Eqs (8) to (10)).

$$\text{Friability (Y1)} = 0.80 - 0.75X_1 + 0.44X^2_1 \qquad \text{Eq (8)}$$

$$\text{Disintegration time (Y2)} = 20.08 + 8.37X_1 - 12.89X_2 \qquad \text{Eq (9)}$$

$$\text{Wetting time (Y3)} = 32.86 + 5.93X_1 - 22.81X_2 + 7.82X^2_2 \qquad \text{Eq (10)}$$

X$_1$ and X$_2$ are CF and crospovidone concentration respectively.

The relative change of a variable is directly related to the size of its regression coefficient. This means that if the model parameters have either a large positive or negative value the corresponding variable has a large influence on the response (s) [55]. It specifies the amount of change in the outcome variable when one predictor variable is changed by one unit (E.g., from 0 to +1) while holding the other factor(s) constant. X$_1$ and X$_2$ represent the average result if changing one variable at a time from its low level to its high level. Coefficients with more than one factor term (X$_1$X$_2$) represent the interaction terms and shows the response variable changes when 2 variables are simultaneously changed. Coefficients with higher order terms indicate the quadratic (non-linear) nature of the relationship [58, 59].

**Table 17. Summary of ANOVA results of the reduced quadratic model for wetting time.**

| Source | Sum of squares | DF | Mean square | F-value | P-value Prob>F | Reamrk |
|---|---|---|---|---|---|---|
| Model | 4878.17 | 3 | 1626.06 | 132.98 | < 0.0001 | Significant |
| X$_1$ | 281.35 | 1 | 281.35 | 23.01 | 0.0010 | |
| X$_2$ | 4164.09 | 1 | 4164.09 | 340.53 | < 0.0001 | |
| X$^2_2$ | 432.73 | 1 | 432.73 | 35.39 | 0.0002 | |
| Residual | 110.05 | 9 | 12.23 | | | |
| Lack of fit | 91.08 | 5 | 18.22 | 3.84 | 0.1084 | Insignificant |
| Pure error | 18.98 | 4 | 4.74 | | | |
| Core total | 4988.22 | 12 | | | | |

**Table 18. The R-squared values of the reduced quadratic model for wetting time.**

| R-Squared | Adj R-Squared | Pred R-Squared | PRESS | Adeq Precision |
|-----------|---------------|----------------|-------|----------------|
| 0.9779 | 0.9706 | 0.9282 | 358.22 | 35.484 |

For the regression coefficients, both the magnitude and sign are important: the magnitude shows the strength of influence, whereas, the sign indicates the direction of the effect on the response variable (s) [50].

As shown from (Eqs 8 to 10) two of the responses (DT and WT) were directly affected by CF. This is to mean that both Eqs (9) and (10) have a positive coefficient for factor ($X_1$) and this implies as CF increases the DT and the WT also increases whereas in Eq (8) the coefficient is negative, indicating the reduction of percentage friability as CF increases. Since the magnitude is higher for DT (+8.37) than WT which is +5.93 and friability (-0.75), it can be concluded that CF has stronger effect on DT. The super disintegrant concentration ($X_2$) has a strong indirect effect for both DT (-12.89) and WT (-22.81).

Significance of quadratic terms could signal that the relation is non-linear. A positive quadratic term could suggest that the relation is exponential. The small positive coefficient of the Quadratic terms ($X^2_1$, +0.44), shows a direct weak relationship with tablet friability. The effect of the quadratic term of the superdisintegrant concentration ($X^2_2$) on WT is strong with a positive sign +7.82 which implies a direct relationship.

**3.3.5. Contour and response surface plot analysis.** The relationship between independent and dependent variables was graphically represented by 3D response surface and 2D contour plots generated by their respective model (Figs 11 to 13). These plots are very useful to show interaction effects of the factors on the responses and are able to show effects of two factors on the response at a time [60].

The above figures, (Fig 11A and 11B) depict the combined effect of CF and crospovidone concentration on tablet friability. The contour plot and surface plot indicates the superdisintegrant concentration does not affect tablet friability significantly. At different concentration of crospovidone maintaining CF constant, the variation on friability was very minimal. But the effect of CF is vivid. As CF increases the friability value decreases dramatically. Besides, there is a slight curvature on the surface and contour plot which showed significant effect quadratic term on friability. This was further confirmed by the ANOVA result of P value < 0.0001 for both $X_1$ and $X^2_1$ and regression coefficients in the mathematical model generated for friability (Eq 8).

Fig 12 depicts the effect of CF and crospovidone concentration on DT. As displayed from the contour and surface plot, the DT of co-trimoxazole ODT was found to be affected by both factors. The graphs showed decrement of DT as a function of super disintegrant concentration increment and decrement of CF. This was further confirmed by the regression equation which

**Table 19. Summary of ANOVA results of response surface models for drug disintegration time.**

| Response | Source | Sum of squares | Df | Mean Square | F-value | P-value | Remark |
|----------|--------|----------------|----|-------------|---------|---------|--------|
| Disintegration time | Model | 1889.48 | 2 | 944.74 | 13.43 | 0.0015 | significant |
| | Compression force ($X_1$) | 560.1 | 1 | 560.14 | 7.96 | 0.0181 | significant |
| | Crospovidone (X2) | 1329.34 | 1 | 1329.34 | 18.90 | 0.0015 | significant |
| | Residual | 703.44 | 10 | 70.34 | | | |
| | Lack of fit | 614.24 | 6 | 102.37 | 4.59 | 0.0809 | Not-significant |
| | Pure error | 89.20 | 4 | 22.30 | | | |
| | Core total | 2592.92 | 12 | | | | |

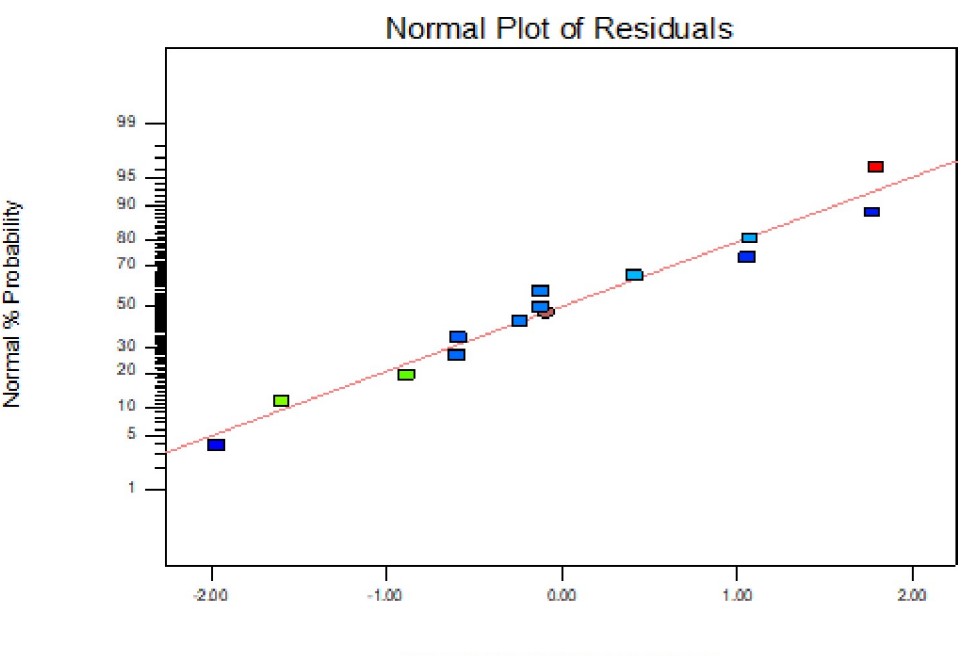

**Fig 5. Normal probability plot of residuals for friability.**

was +8.37 for $X_1$ and -12.89 for $X_2$. Since the magnitude is higher for $X_2$ (-12.89) the effect on disintegration was higher for crospovidone concentration than CF (+8.37). Since the graphs lie almost on a straight line, without any curvature, it shows a linear relationship.

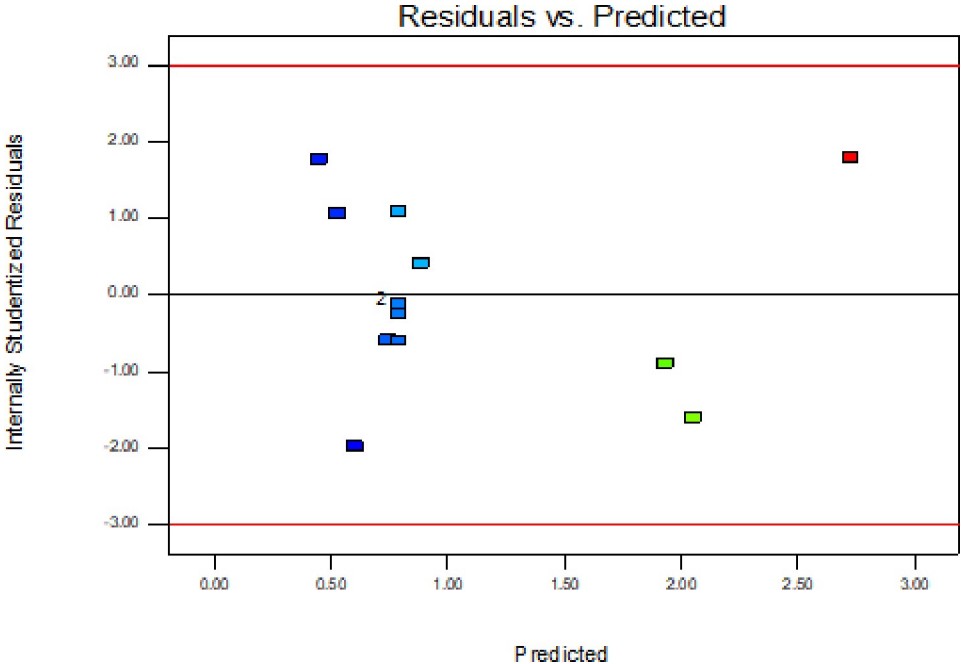

**Fig 6. Plots of the residuals against predicted response for friability.**

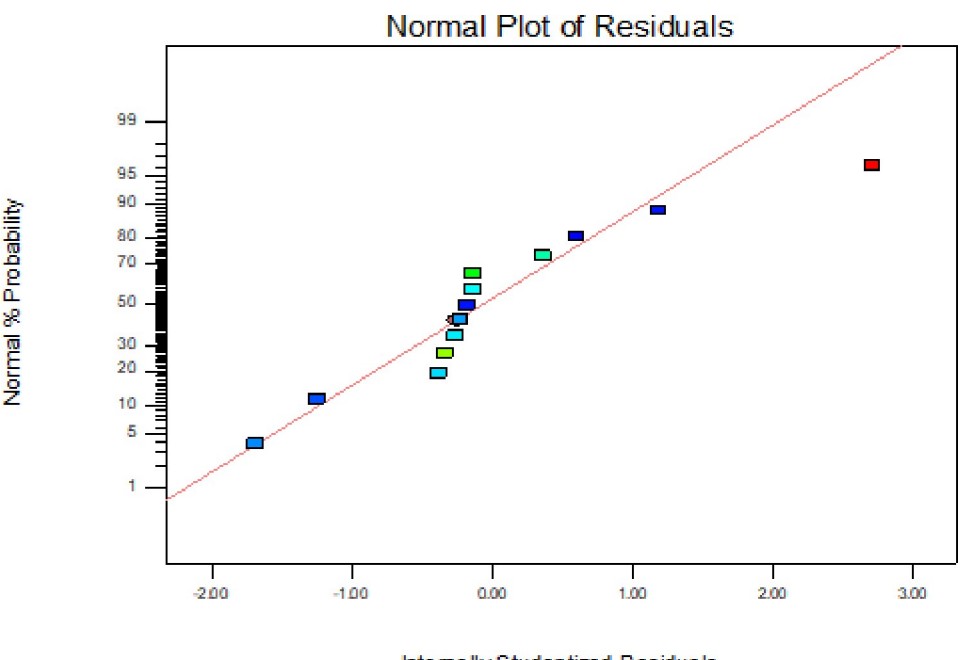

**Fig 7. Normal probability plot of residuals for disintegration time.**

As depicted above by the surface and contour plots (Fig 13A and 13B) the WT decreases as the superdisintegrant concentration increases and as the CF decreases. But at higher super disintegrant concentration there was a slight difference in the WT of tablets even at lower and higher CF. Yet, when the two factors are compared, the supper disintegrant concentration has

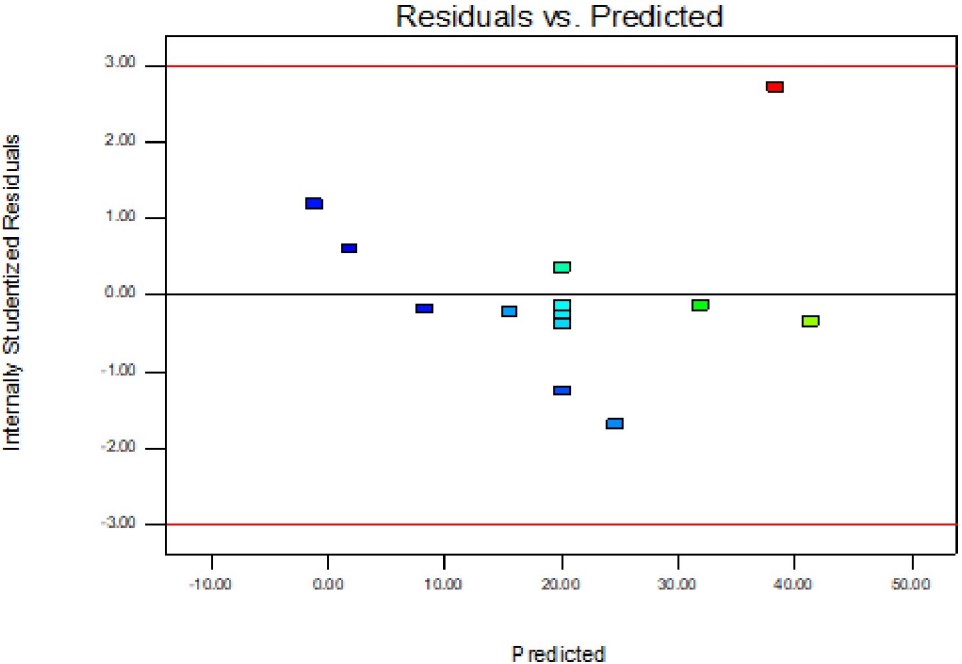

**Fig 8. Plots of the residuals against predicted response for disintegration time.**

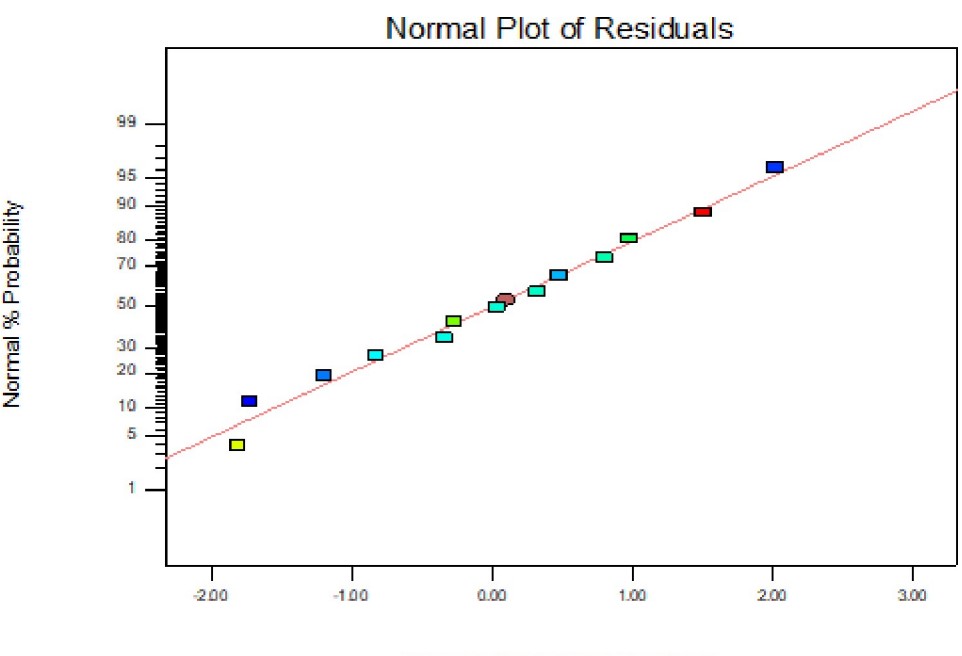

**Fig 9. Normal probability plot of residuals for wetting time.**

greater effect on wetting time than CF which can be further confirmed by the regression coefficients which were found to be -22.8 and +5.93 respectively. According to the plots, the most suitable conditions for minimum WT were found to be at the lowest CF (coded $X_1 = -1$) and at the highest concentration of superdisintegrant (coded $X_2 = +1$).

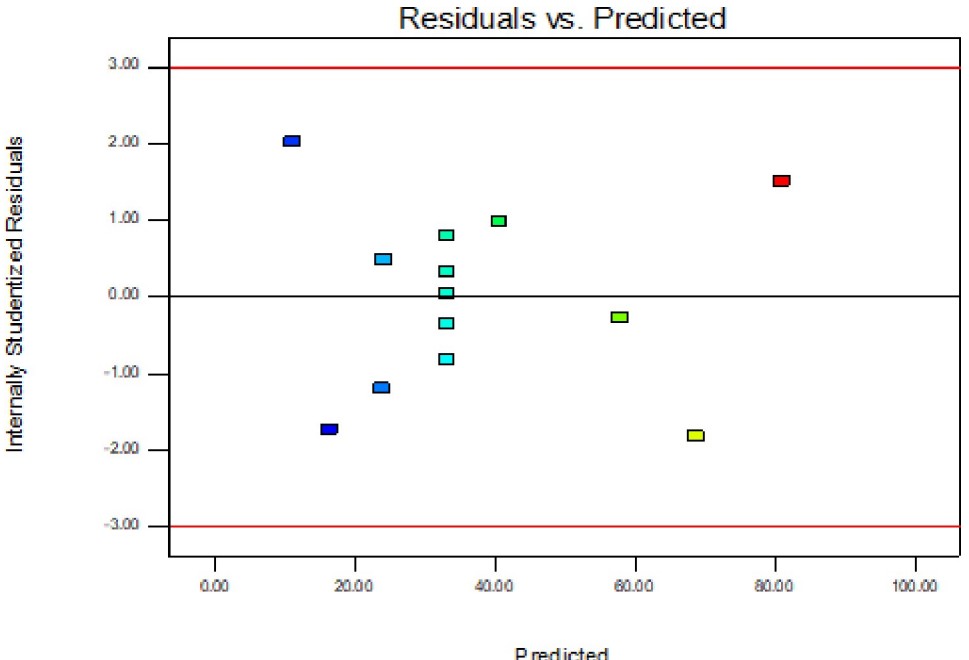

**Fig 10. Plots of the residuals against predicted response for wetting time.**

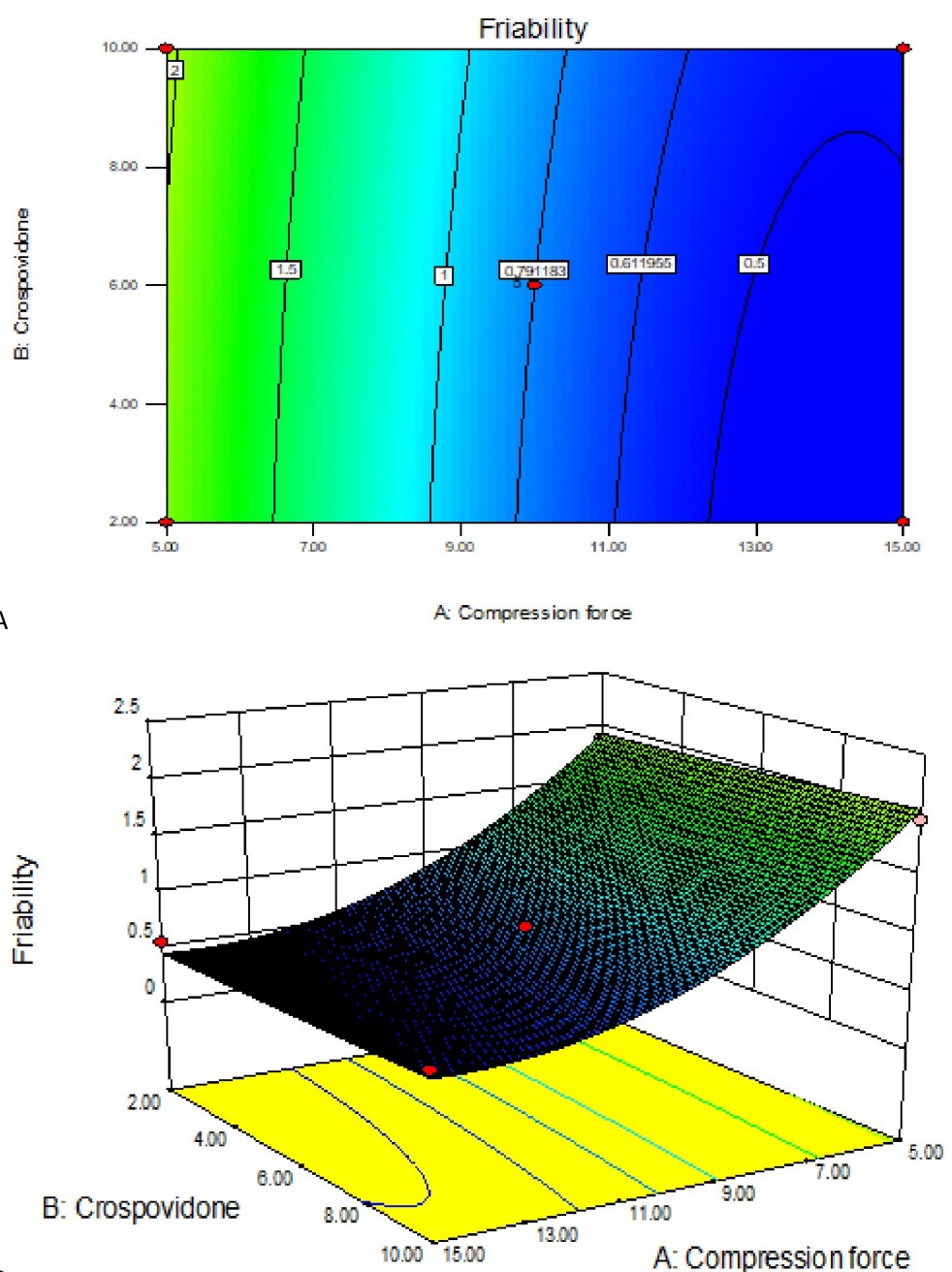

**Fig 11. Contour and response surface plot (A&B) of polymer percentage and compression force on tablet friability.**

**3.3.6. Simultaneous optimization of response variables.** After generating the polynomial equations that relate the dependent and independent variables, the formulation was optimized for all the three responses simultaneously. The final optimal experimental parameters were obtained using both numerical and graphical optimization techniques. The software is capable of generating models and calculate the optimum conditions by making some adjustments or compromise to get a combination of factor levels that jointly optimize a set of responses by satisfying the requirements (i.e. optimization criteria) for each of them i.e.

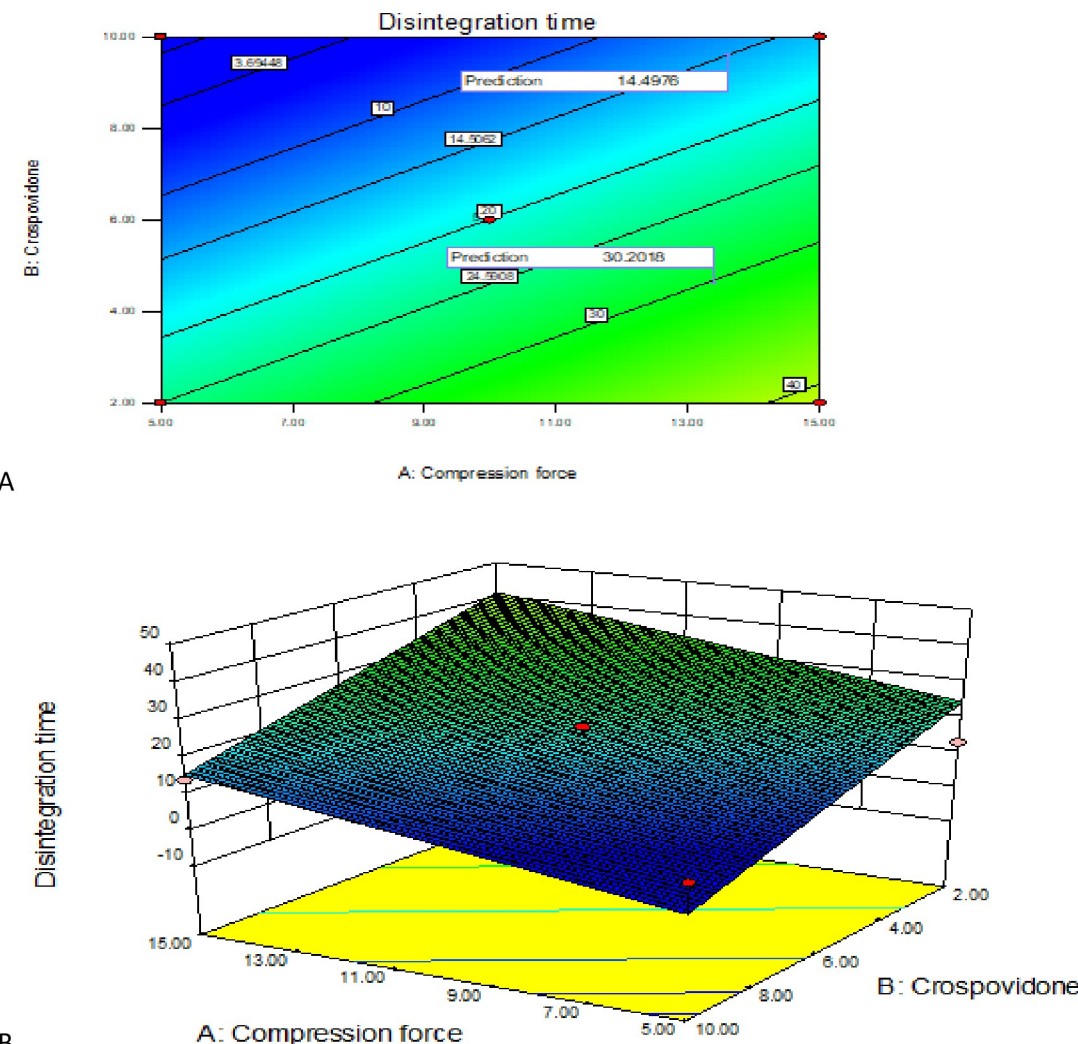

**Fig 12. Contour and response surface plot (A&B) of compression force and polymer percentage on tablet disintegration time.**

multiple response optimization [61]. The constraints on the response variables the was in range of 0.5 to 0.8% for friability, 5 to 15 seconds for DT and 12.3–25 seconds for WT.

*3.3.6.1. Numerical optimization.* In numerical optimization, the desired goals for each factor and response can be chosen. The constraints or defined criteria for factors and responses during numerical and graphical optimization are presented in Table 20. The numerical optimization feature in the design expert package finds one point or more in the factors domain that would maximize this objective function. The process involves combining the goals into an overall desirability function (D).

The desirability approach is recommended due to its simplicity, availability in the software and the flexibility it gives in weighting and giving importance for individual response [62]. The desirability function transforms each estimated response, Yi, into a unit less utility bounded by $0 < di < 1$, where a higher di value indicates that response value Yi is more desirable, and when di = 0 it means a completely undesired response. The variance of desirability value between 0 and 1 depends on the proximity of the outputs towards the target [63, 64].

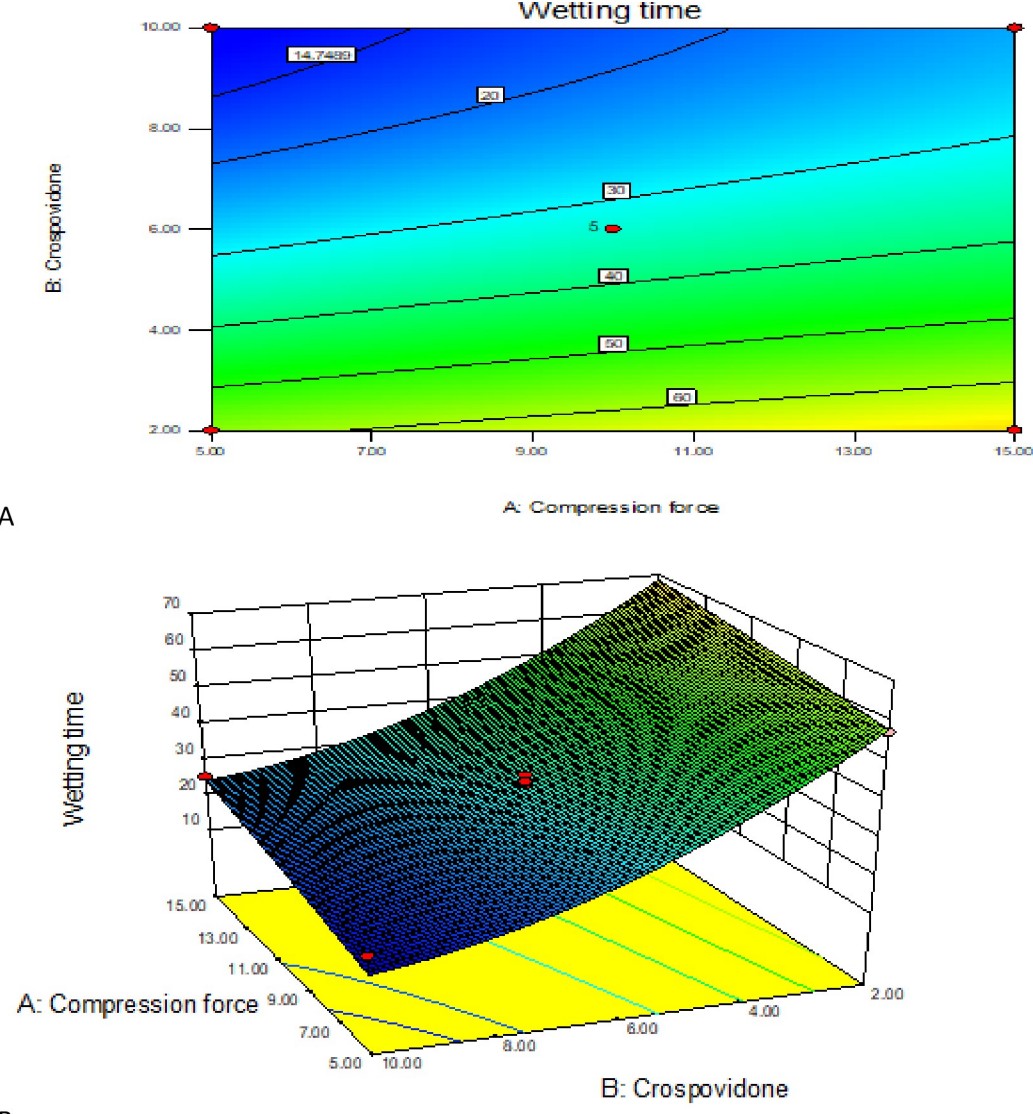

**Fig 13. Contour and response surface plot (A&B) of compression force and polymer percentage of tablet wetting time.**

The overall desirability function $D(x)$ for ($X_1$, $X_2$) is displayed both in ramp and 3D plot in Figs 14 and 15 respectively.

*3.3.6.2. Graphical optimization.* In a graphical optimization with multiple responses, the software defines regions where requirements simultaneously meet the proposed criteria. This could be visualized graphically by superimposing the contours of the response surfaces in an overlay plot. Then, a visual search or selection for the best compromise becomes possible. The graphical optimization displays the area of feasible response values in the factor space and the regions that do fit the optimization criteria would be shaded [56].

Fig 16 depicts the overlay plot by which the yellow shaded area represents the area satisfying the proposed criteria. The values showed by the flag on the shaded area were selected as representative of the optimized area. The values were 8.60% for the percentage of crospovidone

**Table 20. Constraints used for the variables during numerical and graphical optimization.**

| Factor constraints | Level | | | | |
|---|---|---|---|---|---|
| Factor | Low | High | | | |
| Compression force | 5KN | 15KN | | | |
| Crospovidone conc. | 2% | 10% | | | |
| Response constraints | Goal | | | | |
| Response | | | Lower limit | Upper limit | Importance |
| Friability | In range | | 0.5 | 0.8 | 5 |
| Disintegration time | In range | | 5 | 15 | 5 |
| Wetting time | In range | | 12.3 | 25 | 3 |

concentration and 11.25KN for CF to get a predicted percentage friability of 0.666%, a DT of 13.794 seconds and 23.186 seconds for the WT with a desirability of 1.00.

**3.3.7. Confirmation test.** To confirm the validity of the obtained predicted optimal point, confirmation experiments were carried out at the optimal combinations of the factors in triplicate ($X_1$ = 11.25KN and $X_2$ = 8.60%). Tablets were evaluated for the outcome variables, (Friability, DT and WT) and the experimental outcomes were compared with the model predicted responses (Table 21). The percent error for each response was investigated for validation of the experiments. Errors between predicted and actual values were calculated according to equation (Eq 11)

$$\%\text{Error} = \frac{(Actual\ value - Predicted\ value) * 100}{Actual\ value} \qquad \text{Eq (11)}$$

Accordingly, if the percent error is found within 5 percent deviation from the actual result; we confirm the validity of the response model and we can say optimization processes is capable and reliable to optimize the response variables [33, 65].

As shown in Table 21, the predicted values, experimental results and the percentage error values are presented. The values of percentage errors for all the response variables were below 5%, confirming that the experimental values of the optimized formulations showed good agreement with the predicted values.

*3.3.7.1. Powder characteristics of the three optimized formulations.* After mixing all the excipients and the active ingredients according to the final optimized formula, each of the three

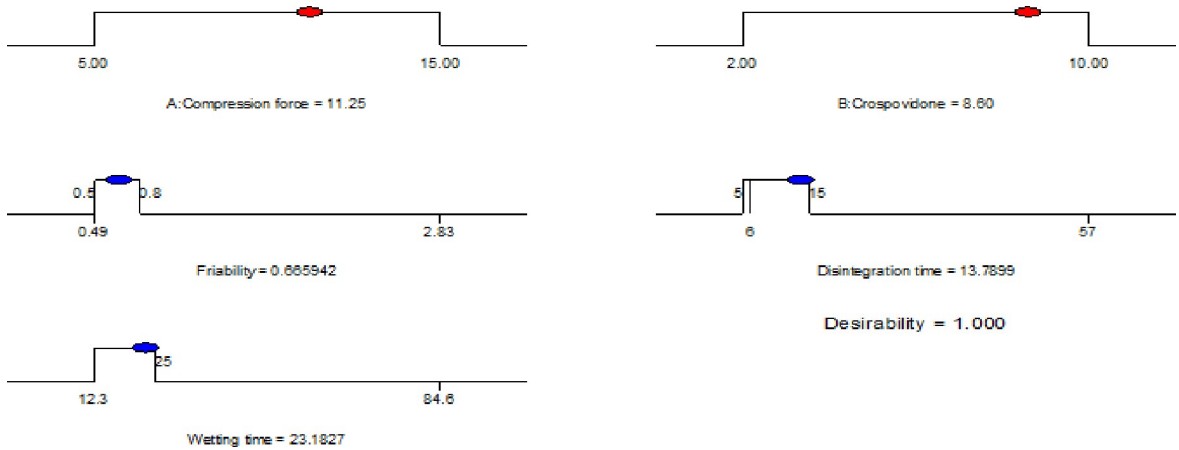

**Fig 14. Numerical optimization results of predicted optimum values and the corresponding levels of factors.**

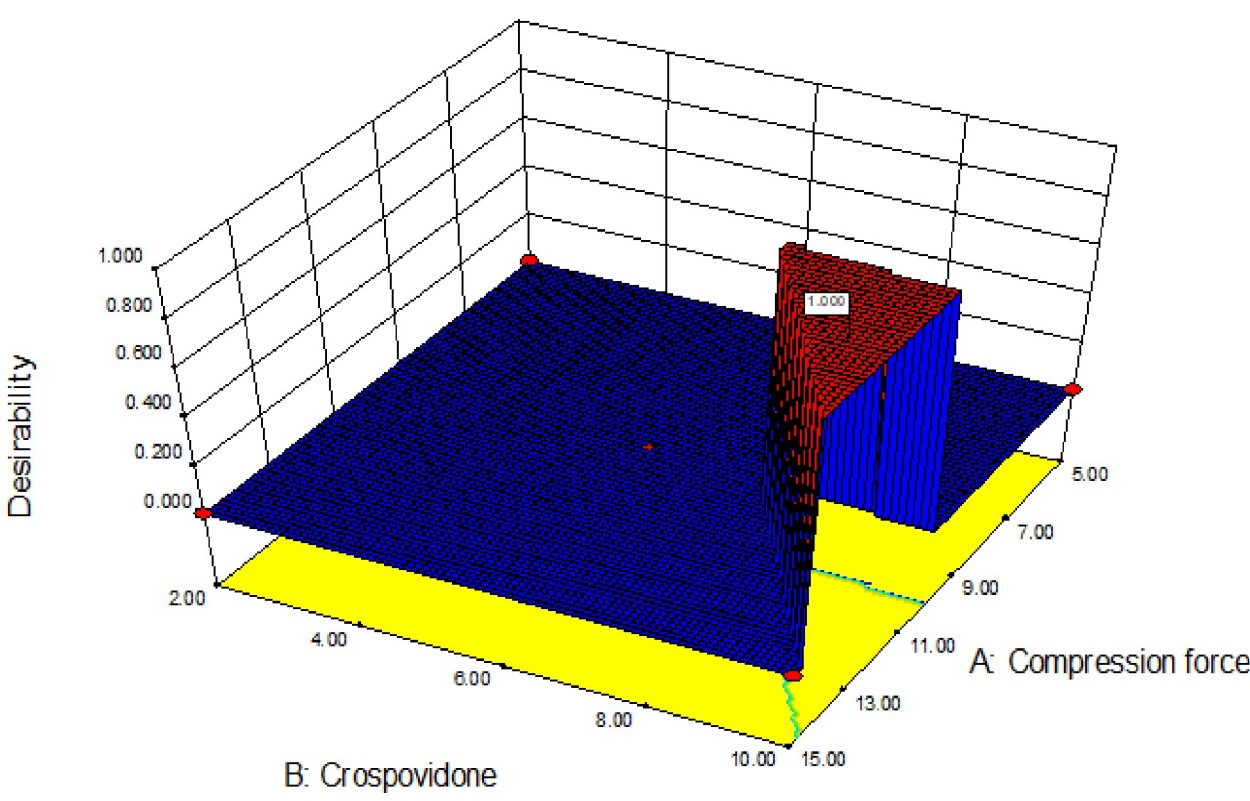

**Fig 15. The 3D plot of the overall desirability function.**

batches powder blend was characterized for their flow property and the result is depicted in (Table 22) which showed good to excellent flow.

**3.3.7.2. Evaluation of the optimized co-trimoxazole ODT tablets.** As displayed in Table 23, tablets of all the 3 formulations were evaluated for thickness, diameter, hardness, and

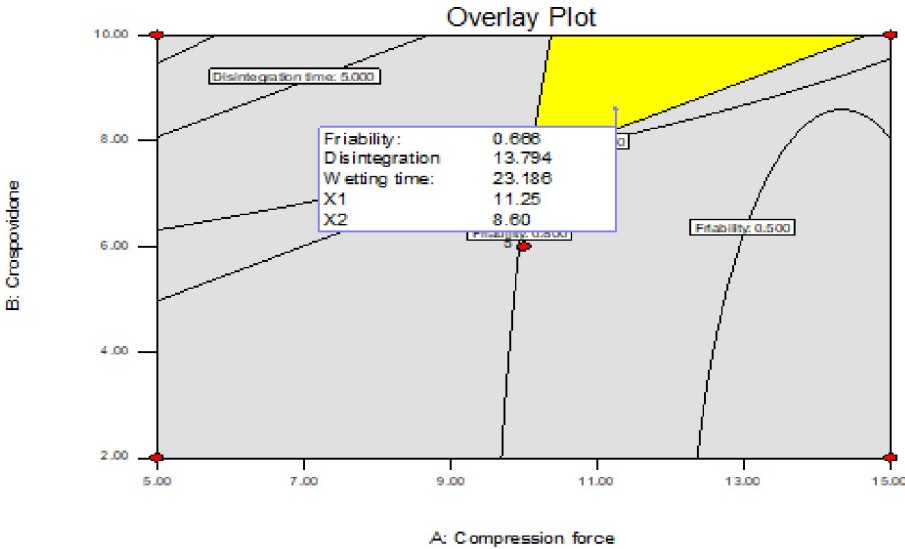

**Fig 16. Graphical representation of the contours of the response surfaces in an overlay plot.**

**Table 21. Response values of predicted, actual obtained experimentally and percentage error.**

| Response variables | Predicted value | Actual value | % Error |
|---|---|---|---|
| Friability | 0.666% | 0.68% | 2.06% |
| Disintegration time | 13.79 Seconds | 14.33 Seconds | 3.77% |
| Wetting time | 23.19 Seconds | 24.00 Seconds | 3.38% |

**Table 22. Physicochemical characterization of powder blends of the optimized formulations.**

| Parameters | Experimental values |
|---|---|
| Bulk density (g/cm$^3$) | 0.613±0.005 |
| Tapped density (g/cm$^3$) | 0.71±0.015 |
| Carr's Index % | 14.003±1.11 |
| Hausner's ratio | 1.163±0.015 |
| Angle of repose (°) | 27.11±0.89 |

**Table 23. Characteristic properties of optimized co-trimoxazole ODT.**

| Formulation code | Thickness | Diameter | Hardness | Assay (TMP) | Assay(SMX) |
|---|---|---|---|---|---|
| F1 | 2.76± 0.03 | 9.93±0.014 | 7.3±0.12 | 96.85±0.035 | 98.53±0.06 |
| F2 | 2.78±0.01 | 9.93±0.015 | 7.01±0.28 | 97.48±0.02 | 99.3±0.00 |
| F3 | 2.71±0.05 | 9.95±0.018 | 7.46±0.16 | 96.4±0.14 | 98.1±0.1 |

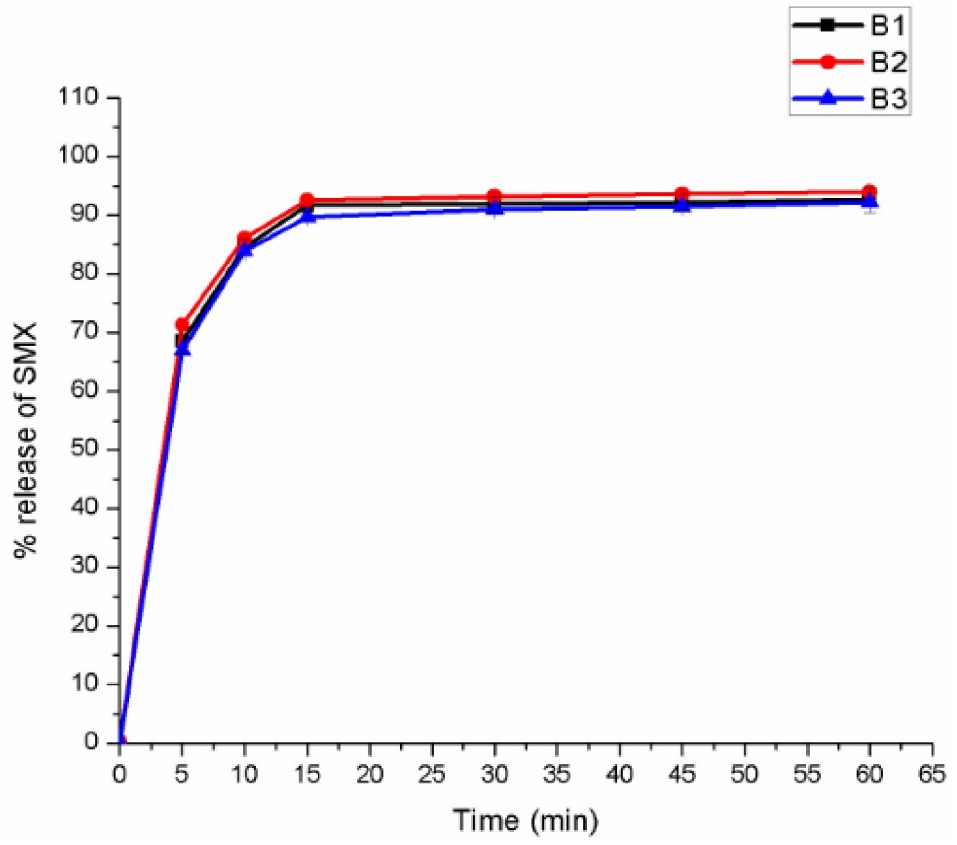

**Fig 17. Percentage drug release profile of SMX in the three optimized batches of co-trimoxazole ODTs.**

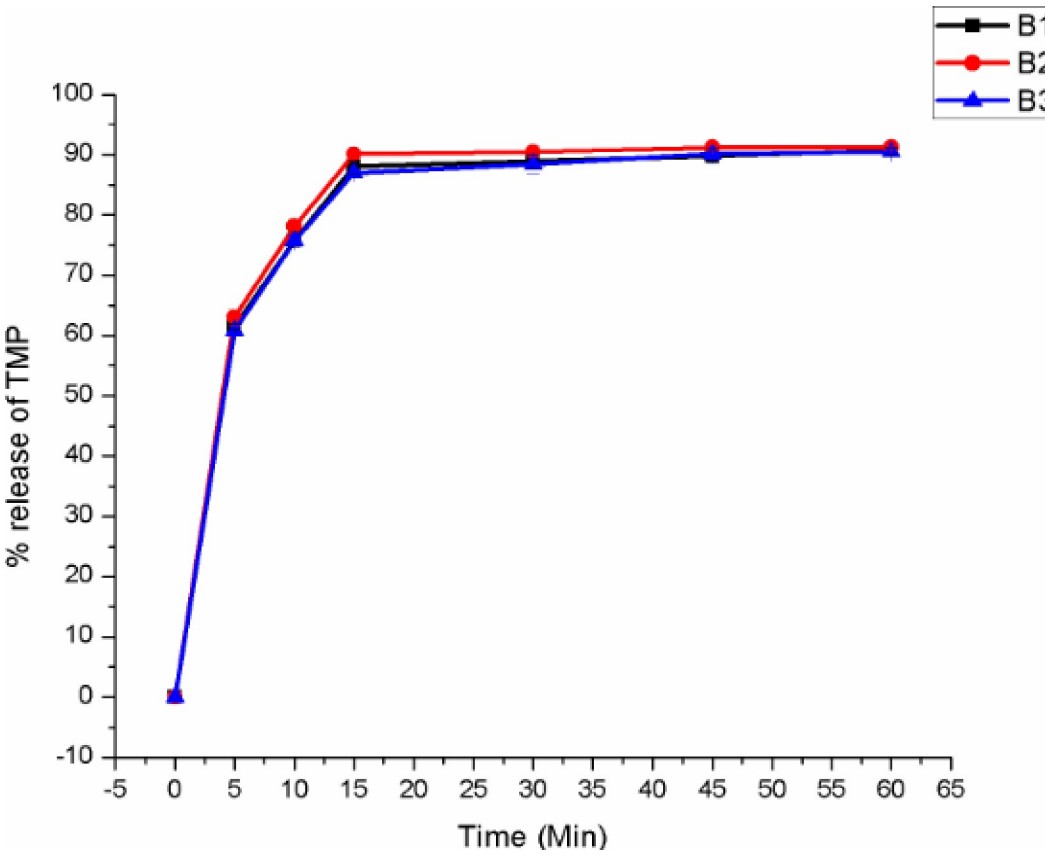

**Fig 18. Percentage drug release profile of TMP in the three optimized batches of co-trimoxazole ODTs.**

drug content and all fulfilled the pharmacopoeial specification. The assay results of the optimized batches were found within a narrow range around the label claim evidenced by a small standard deviation, which ensures the consistency of the dosage units.

In addition to characterizing the optimized formulation for the above parameters, the release profile of the drug was also determined, Figs 17 and 18 and Table 24. The results showed that the optimized co-trimoxazole ODTs exhibited good *in vitro* drug release profile.

As displayed from the dissolution test results, it can be observed that all the three batches showed a mean release of more than 70% before 60 min which satisfies the tolerance limit. These results showed that the optimized CTX ODT had good *in vitro* drug release profile.

## 4. Conclusion

Preparation of taste masked orally disintegrating co-trimoxazole tablet was possible by the DC method using crospovidone as a super disintegrant. The taste masking was successful with a

**Table 24. Assay and cumulative release of the optimized formula.**

| Formulation code | Assay ± RSD (%) | | Dissolution ± RSD (%) | |
|---|---|---|---|---|
| | TMP | SMX | TMP | SMX |
| F1 | 96.85±0.04 | 98.53±0.06 | 90.81±0.43 | 92.69±0.62 |
| F2 | 97.48±0.02 | 99.3±0.00 | 91.4±0.56 | 93.95±0.35 |
| F3 | 96.4±0.14 | 98.1±0.1 | 90.04±0.33 | 92.18±0.59 |

combination of sweetening agent, saccharin sodium and solid dispersion with the solvent evaporation technique. All the formulation blends showed good flow properties such as angle of repose, bulk density, tapped density which revealed that they could be prepared by DC method without flow problem. Besides the prepared tablets showed good post compression parameters.

Co-trimoxazole ODT formulation has been developed and optimized successfully using central composite design. The method was found effective for estimating the effect of two main independent variables (CF and crospovidone concentration) by using polynomial equation and surface plots. Optimization of the three response variables was possible by using both numerical and graphical optimization and the predicted optimal conditions were confirmed experimentally and were found to be in good agreement within 5% of the predicted responses.

The results of this study showed that co-trimoxazole ODT had rapid disintegration, optimum percentage friability which was less than 1% and enhanced dissolution profiles. The successful formulation of co-trimoxazole ODT can solve difficulty of swallowing of conventional tablets for some group of patients which are unable to swallow solid oral dosage form. In addition, the cost of production of CTX suspension, which requires sterile condition, transportation cost due to bulkiness of bottles could also be avoided.

## Supporting information

**S1 Material.**
(RAR)

## Acknowledgments

I would like to express my deepest gratitude to Addis Pharmaceutical Factory SC. for their gift of all the excipients, API and reference standard of TMP and SMX and Food, Medicine and Health care Administration and Control Authority of Northern region (FMHACA) for allowing me to use HPLC in their lab.

## Author Contributions

**Conceptualization:** Chernet Tafere, Zewdu Yilma, Solomon Abrha, Adane Yehualaw.

**Formal analysis:** Chernet Tafere.

**Funding acquisition:** Chernet Tafere.

**Investigation:** Chernet Tafere.

**Methodology:** Chernet Tafere.

**Supervision:** Zewdu Yilma, Solomon Abrha, Adane Yehualaw.

**Writing – original draft:** Chernet Tafere.

**Writing – review & editing:** Chernet Tafere.

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
