## [Decision Letter · Decision Letter 0]

26 Oct 2020

PONE-D-20-30735

Formulation, in vitro Characterization and Optimization of Taste-Masked Orally Disintegrating Co-trimoxazole Tablet by Direct Compression

PLOS ONE

Dear Dr. Tafere,

Thank you for submitting your manuscript to PLOS ONE. After careful consideration, we feel that it has merit but does not fully meet PLOS ONE’s publication criteria as it currently stands. Therefore, we invite you to submit a revised version of the manuscript that addresses the points raised during the review process.

Based on the recommendation of reviewer, the manuscript needs extensive revision before consider for publication.

We look forward to receiving your revised manuscript.

Kind regards,

Girish Sailor

Academic Editor

PLOS ONE

Journal Requirements:

2. Please provide additional details regarding participant consent. In the ethics statement in the Methods and online submission information, please ensure that you have specified whether consent was informed.

3. In the Methods section, please provide the product number and any lot numbers of the materials purchased from chemical companies for your study.

4. In your Methods section, please provide additional information about the participant recruitment method and the demographic details of your participants. Please ensure you have provided sufficient details to replicate the analyses such as: a) the recruitment date range (month and year),  b) a table of relevant demographic details, c) a statement as to whether your sample can be considered representative of a larger population, d) a description of how participants were recruited, and e) descriptions of where participants were recruited and where the research took place.

5. In the Methods section, please provide detailed methods of taste masking by solid dispersion used in your study, including but not limited to preparation, solvents and ratios used, temperature and time, yield, quality, characterization, other ingredients/compounds included in the final extract, and confirmation the targeted compound was extracted.

6. Please provide a sample size and power calculation in the Methods, or discuss the reasons for not performing one before study initiation.

7.Thank you for stating the following in the Acknowledgments Section of your manuscript:

[My great acknowledgement also goes to Adigrat University for sponsoring my postgraduate

study and Mekelle University for sponsoring my thesis research]

 [The funders had no role in study design, data collection and analysis, decision to publish, or preparation of the manuscript.]

8. Please amend your list of authors on the manuscript to ensure that each author is linked to an affiliation. Authors’ affiliations should reflect the institution where the work was done (if authors moved subsequently, you can also list the new affiliation stating “current affiliation:….” as necessary).

9. Please amend either the abstract on the online submission form (via Edit Submission) or the abstract in the manuscript so that they are identical.

10. Please ensure that you refer to Figure 3.14 and 3.15 in your text as, if accepted, production will need this reference to link the reader to the figure.

11. Please include a copy of Table 4.4, 4.10, 4.11, 4.14 and 4.16 which you refer to in your text on page 17, 24, 26, 28 and 29.

12. We note you have included a table to which you do not refer in the text of your manuscript. Please ensure that you refer to Table 3.4, 3.10, 3.11, 3.13, 3.14, 3.16 and 3.17 in your text; if accepted, production will need this reference to link the reader to the Table.

13. Please upload a copy of Figure 4.14 and 4.15, to which you refer in your text on page 33. If the figure is no longer to be included as part of the submission please remove all reference to it within the text.

14. Please include a caption for figure Figure 4.14 and 4.15.

Reviewers' comments:

Reviewer's Responses to Questions

**Comments to the Author**

1. Is the manuscript technically sound, and do the data support the conclusions?

Reviewer #1: Partly

Reviewer #2: Partly

2. Has the statistical analysis been performed appropriately and rigorously? 

Reviewer #1: I Don't Know

Reviewer #2: I Don't Know

3. Have the authors made all data underlying the findings in their manuscript fully available?

Reviewer #1: Yes

Reviewer #2: Yes

4. Is the manuscript presented in an intelligible fashion and written in standard English?

Reviewer #1: Yes

Reviewer #2: No

5. Review Comments to the Author

Reviewer #1: abbreviations used in abstract are confusing and must be written in full form (TMP & KN).

Referencing in text is not proper. You have used numbering but many times you have written name of authors with year in text.

Repeated use of make and model of instruments must be avoided.

Why used 0.1N HCl as dissolution medium.

Where drug dissolution will take place, in mouth or stomach.

Readings of hardness and friability are not showing good correlation.

Reviewer #2: The premise of the paper is ok. But the authors need to take a hard look at the paper and make it crisp and to the point. Review comments have been added in the paper.

The authors are advised to get the paper copy edited.

6. PLOS authors have the option to publish the peer review history of their article (what does this mean?). If published, this will include your full peer review and any attached files.

Reviewer #1: No

Reviewer #2: No

---

## [Author Response · Author response to Decision Letter 0]

4 Jan 2021

Point by point response 

Reviewer #1 

Comments #1: Please ensure that your manuscript meets PLOS ONE's style requirements, including those for file naming. 

Authors’ response #1: Dear Doctor, Thank you, the correction is made accordingly.

Comment #2: Please provide additional details regarding participant consent. In the ethics statement in the Methods and online submission information, please ensure that you have specified whether consent was informed.

Authors’ response #2: Thank you Doctor once again. With all due respect, I have tried to mention that informed consent was taken and have been specified appropriately on the method section. But I have added some sentence that will strengthen that consent was informed.

Comment #3: In the Methods section, please provide the product number and any lot numbers of the materials purchased from chemical companies for your study.

Authors’ response #3: Dear Doctor, most of the ingredients product number is written.

Comment #4: In your Methods section, please provide additional information about the participant recruitment method and the demographic details of your participants. Please ensure you have provided sufficient details to replicate the analyses such as: a) the recruitment date range (month and year), b) a table of relevant demographic details, c) a statement as to whether your sample can be considered representative of a larger population, d) a description of how participants were recruited, and e) descriptions of where participants were recruited and where the research took place.

Authors’ response #4: Dear doctor, I have tried to state the demographic details in the section of Determination of taste threshold value. I have stated that they all are above 18, literate and non-smoking. Besides, I have added a statement of where the participants were recruited that is (Mekelle University) and the research was done in Mekelle Ethiopia for the partial fulfillment of a master’s degree in pharmaceutics. The sample size might not be considered representative of a larger population, but I have used an already used method which was published in American-Eurasian Journal of Scientific Research. (DOI: 10.5829/idosi.aejsr.2012.7.2.1507)

Comment #5: In the Methods section, please provide detailed methods of taste masking by solid dispersion used in your study, including but not limited to preparation, solvents and ratios used, temperature and time, yield, quality, characterization, other ingredients/compounds included in the final extract, and confirmation the targeted compound was extracted.

Authors’ response #5: Thanks Doctor for your great review of this paper, I see your point and it make sense if you believe that there is a target compound to be extracted. But, this research doesn’t involve any extraction process. However, the research uses a carrier molecule (Eudragit E100) which is pH sensitive polymer which solubilizes at a lower pH (stomach) but not at a higher pH like that of saliva’s pH to conceal the bitter test of the drug at salivary pH so that not sensed by our tongue. Whether there is drug release from the taste masked solid dispersion (TMP- Eudragit E100 complex) was determined using in-vivo and in vitro test evaluation at salivary pH. 

Comment #6: Please provide a sample size and power calculation in the Methods, or discuss the reasons for not performing one before study initiation.

Authors’ response #6: Dear Doctor, I have made the taste threshold determination according to the method described by (Kulkarni AP, Khedkar AB, Lahotib SR., 2012) who did a research on Development of Oral Disintegrating Tablet of Rizatriptan Benzoate with Inhibited Bitter Taste. Since I didn’t develop my own method instead used an already published and validated method, I don’t think that’s necessary to do sample size calculation.

Comment #7: Thank you for stating the following in the Acknowledgments Section of your manuscript. Please remove any funding-related text from the manuscript and let us know how you would like to update your Funding Statement.

Authors’ response #7: Thanks, the correction is made.

Reviewer #2 

Comments #8: Please amend your list of authors on the manuscript to ensure that each author is linked to an affiliation. Authors’ affiliations should reflect the institution where the work was done (if authors moved subsequently, you can also list the new affiliation stating “current affiliation:….” as necessary).

Authors’ response #8: Thank you Doctor, I have made the correction.

Comments #9: Please amend either the abstract on the online submission form (via Edit Submission) or the abstract in the manuscript so that they are identical.

Authors’ response #9: Thank you Doctor, it’s done.

Comments #10: Please ensure that you refer to Figure 3.14 and 3.15 in your text as, if accepted, production will need this reference to link the reader to the figure.

Authors’ response #10: Figure 3.14 and 3.15 have been mistakenly referred on the text as Figure 4.14 and Figure 4.15 and it’s now corrected.

Comments #11: Please include a copy of Table 4.4, 4.10, 4.11, 4.14 and 4.16 which you refer to in your text on page 17, 24, 26, 28 and 29.

Authors’ response #11: They are all corrected as Table 3.4, 3.10, 3.11, 3.14, 3.16.

Comments #12: We note you have included a table to which you do not refer in the text of your manuscript. Please ensure that you refer to Table 3.4, 3.10, 3.11, 3.13, 3.14, 3.16 and 3.17 in your text; if accepted, production will need this reference to link the reader to the Table.

Authors’ response #12: Thanks doctor, done as requested.

Comment #13: Please upload a copy of Figure 4.14 and 4.15, to which you refer in your text on page 33. If the figure is no longer to be included as part of the submission please remove all reference to it within the text.

Authors’ response #13: thanks doctor for your deep review. Here’s what happened. The objective had been given Heading 2 when the thesis was done. But when the manuscript was sent for publication to Plos one, the objective was combined as part of the introduction. This resulted in the reduction of Heading 4 of Result and Discussion part to Heading 3, and in the meantime some of tables and figures were forgotten as they had been. They are now corrected as Figure 3.14 and 3.15

Comment #14: Please include a caption for Figure 4.14 and 4.15.

Authors’ response #14: Dear Doctor, they were mistakenly referred as Figure 4.14 and 4.15, they are now corrected as Figure 3.14 and 3.15

General comments

Comment: In the introduction part you have asked me why I use direct compression instead of granulation.

Authors’ response: Dear Doctor this is good question. Even though I have tried to justify why I choose DC in the discussion part, I will reply to your question as follows.

DC has low cost of production when compared to granulation as it not uses granulating fluid, has smaller number of production steps. Besides, granulation results in better binding of tablet components which might pose disintegration problem, which is not needed for ODT tablets.

Comment: In the method section you have asked me when the SMX was added.

Authors’ response: All the ingredients, including the appropriately stored, solid dispersed Trimethoprim and Eudragit complex, SMX and the sweetener Saccharin sodium, were allowed to pass through 45 meshes or (354 μm) sieve separately. Then they were all mixed in a polybag for five minutes. After this, magnesium stearate and colloidal silicon di oxide were added for lubrication and mixed for 3 minute. Finally each batch was converted into tablet by compression.

Comment: In the result and discussion part you have asked me to mention the final formulation formula somewhere.

Authors’ response: Dear Doctors, The final formulation formula was already been mentioned in Confirmation test section (Heading 3.3.7) as follows:

To confirm the validity of the obtained predicted optimal point, confirmation experiments

were carried out at the optimal combinations of the factors in triplicate (X1= 11.25KN and X2=

8.60%).

Comment: In the result and discussion part you have asked me that p values have not been mentioned in the table 3.4

Authors’ response: Thank you doctors for your amazing view of this paper. Sorry, it was by mistake that I didn’t mention Table 3.5 on the paragraph before the table. The p values were already placed on Table 3.5 and correctly referred on the above paragraph.

---

## [Decision Letter · Decision Letter 1]

25 Jan 2021

Formulation, in vitro Characterization and Optimization of Taste-Masked Orally Disintegrating Co-trimoxazole Tablet by Direct Compression

PONE-D-20-30735R1

Dear Dr. Tafere,

We’re pleased to inform you that your manuscript has been judged scientifically suitable for publication and will be formally accepted for publication once it meets all outstanding technical requirements.

Kind regards,

Girish Sailor

Academic Editor

PLOS ONE

Additional Editor Comments (optional):

Reviewers' comments:

Reviewer's Responses to Questions

**Comments to the Author**

1. If the authors have adequately addressed your comments raised in a previous round of review and you feel that this manuscript is now acceptable for publication, you may indicate that here to bypass the “Comments to the Author” section, enter your conflict of interest statement in the “Confidential to Editor” section, and submit your "Accept" recommendation.

Reviewer #1: All comments have been addressed

Reviewer #2: All comments have been addressed

2. Is the manuscript technically sound, and do the data support the conclusions?

Reviewer #1: Yes

Reviewer #2: Yes

3. Has the statistical analysis been performed appropriately and rigorously? 

Reviewer #1: I Don't Know

Reviewer #2: I Don't Know

4. Have the authors made all data underlying the findings in their manuscript fully available?

Reviewer #1: Yes

Reviewer #2: Yes

5. Is the manuscript presented in an intelligible fashion and written in standard English?

Reviewer #1: Yes

Reviewer #2: Yes

6. Review Comments to the Author

Reviewer #1: (No Response)

Reviewer #2: The authors have tried to address all comments. It would be best if they try to truncate the document by reducing the number of unnecessary tables and figures and definitions. These may be added in the supplementary appendix.

7. PLOS authors have the option to publish the peer review history of their article (what does this mean?). If published, this will include your full peer review and any attached files.

Reviewer #1: No

Reviewer #2: No

---

## [Editor Report · Acceptance letter]

29 Jan 2021

PONE-D-20-30735R1 

Formulation, *in vitro* characterization and optimization of taste-masked orally disintegrating co-trimoxazole tablet by direct compression 

Dear Dr. Tafere:

I'm pleased to inform you that your manuscript has been deemed suitable for publication in PLOS ONE. Congratulations! Your manuscript is now with our production department. 

Kind regards, 

on behalf of

Dr. Girish Sailor 

Academic Editor

PLOS ONE